# COVARIANCES FOR FREE: EXPLOITING MEAN DISTRIBUTIONS FOR FEDERATED LEARNING WITH PRE-TRAINED MODELS

## ABSTRACT

Using pre-trained models has been found to reduce the effect of data heterogeneity and speed up federated learning algorithms. Recent works have investigated the use of first-order statistics and second-order statistics to aggregate local client data distributions at the server and achieve very high performance without any training. In this work we propose a training-free method based on an unbiased estimator of class covariance matrices. Our method, which only uses first-order statistics in the form of class means communicated by clients to the server, incurs only a fraction of the communication costs required by methods based on communicating second-order statistics. We show how these estimated class covariances can be used to initialize a linear classifier, thus exploiting the covariances without actually sharing them. When compared to state-of-the-art methods which also share only class means, our approach improves performance in the range of 4-26% with exactly the same communication cost. Moreover, our method achieves performance competitive or superior to sharing second-order statistics with dramatically less communication overhead. Finally, using our method to initialize classifiers and then performing federated fine-tuning yields better and faster convergence.

## 1 INTRODUCTION

Federated learning (FL) is a widely used paradigm for distributed learning from multiple clients or participants. In FL, each client trains their local model on their private data and then send model updates to a common global server that aggregates this information into a global model. The objective is to learn a global model that performs similarly to a model jointly trained on all the client data. A major concern in existing federated optimization algorithms (McMahan et al., 2017) is the poor performance when the client data is not identically and independently distributed (iid) or when classes are imbalanced between clients (Zhao et al., 2018; Li et al., 2019; Acar et al., 2021; Karimireddy et al., 2020a). Luo et al. (2021) showed that client drift in FL is mainly due to the drift in client classifiers which optimize to the local data distribution, resulting in forgetting knowledge from other clients from previous rounds (Legate et al., 2023b; Caldarola et al., 2022). Another challenge in FL is the partial participation of clients in successive rounds (Li et al., 2019), which becomes particularly acute with large numbers of clients (Ruan et al., 2021; Kairouz et al., 2021). To address these challenges, recent works have focused on algorithms to better tackle data heterogeneity between clients (Luo et al., 2021; Tan et al., 2022b; Legate et al., 2023a; Fanì et al., 2024).

Motivated by results from transfer learning (He et al., 2019), several recent works on FL have studied the impact of using pre-trained models and observe that it can significantly reduce the impact of data heterogeneity (Legate et al., 2023a; Nguyen et al., 2023; Tan et al., 2022b; Chen et al., 2022; Qu et al., 2022; Shysheya et al., 2022; Luo et al., 2021; Tan et al., 2022a). An important finding in several of these works is that sending local class means to the server instead of raw features is more efficient in terms of communication costs, eliminates privacy concerns, and is robust to gradient-based attacks (Chen et al., 2022; Zhu et al., 2019). Tan et al. (2022b) used pre-trained models to compute and then share class means as the representative of each class, and Legate et al. (2023a) showed that aggregating local means into global means and setting them as classifier weights (FedNCM) achieves very good performance without any training. FedNCM incurs very little communication cost and enables stable initialization. Recently, the authors of Fed3R (Fanì et al., 2024) explored the

impact of sharing second-order feature statistics from clients to server to solve the ridge regression problem (Boyd & Vandenberghe, 2004) in federated learning and improves over FedNCM.

Fed3R communicates second-order statistics computed from local features for classifier initialization, and Luo et al. (2021) previously proposed using class means and covariances from all clients for classifier calibration after federated optimization. Although it is evident that exploiting second-order feature statistics results in better and more stable classifiers, it poses new problems. Notably, transferring second-order statistics for high-dimensional features from clients to the server significantly increases the communication overhead and also exposes clients to privacy risks (Luo et al., 2021; Fanì et al., 2024). In order to reap the benefits of second-order client statistics, while at the same time mitigating these risks, we propose Federated learning with COvariances for Free (FedCOF) which only communicates class means from clients to the server. We show that, from just these class means and exploiting the mathematical relationship between

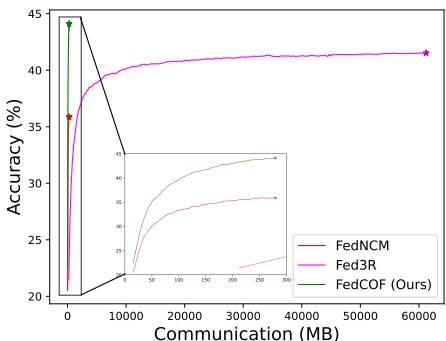

Figure 1: Performance vs. communication cost using pre-trained MobileNetv2 on iNaturalist-Users-120K. Our method (Fed-COF) achieves better accuracy than Fed3R while having the same cost as FedNCM.

the covariance of class means and the class covariance matrices, we can compute an unbiased estimator of global class covariances on the server. Finally, we set the classifier weights in terms of aggregated class means and our estimated class covariances.

In this paper we exploit pre-trained feature extractors and propose a training-free method (FedCOF) that uses the same communication budget as FedNCM while delivering performance comparable to or even superior to Fed3R. FedCOF is based on a provably unbiased estimator of class covariances that requires only class means communicated from clients to the server. We validate our proposed method across several FL benchmarks, including the real-world non-iid iNaturalist-Users-120K, and our results (see Fig. 1) demonstrate that – with only a fraction of the communication costs incurred by methods based on communication of second-order statistics – FedCOF can achieve state-of-the-art results. Furthermore, we show that FedCOF can be used as an initialization for federated optimization methods in order to achieve faster and better convergence.

## 2 RELATED WORK

**Federated learning.** FL is a rapidly-growing field and focuses on neural network training in federated environments (Zhang et al., 2021; Wen et al., 2023). Initial works like FedAvg (McMahan et al., 2017) focuses on the aspect of distributed model training and communication. Later works focus more on non-iid settings, where data among the clients is more heterogeneous (Li et al., 2019; Kairouz et al., 2021; Wang et al., 2021; Li et al., 2021). FedNova (Wang et al., 2020) normalizes local updates before averaging to address objective inconsistency. Scaffold (Karimireddy et al., 2020b) employs control variates to correct drift in local updates. FedProx (Li et al., 2020) introduces a proximal term in local objectives to stabilize the learning process. Reddi et al. (2020) proposed use of adaptive optimization methods, such as Adagrad, Adam and Yogi, at the server side. While CCVR (Luo et al., 2021) proposed a classifier calibration by aggregating class means and covariances from clients, (Li et al., 2023; Dong et al., 2022; Oh et al., 2021; Kim et al., 2024) proposed using a fixed classifier motivated by the neural collapse phenomenon (Papyan et al., 2020). After federated training with fixed classifiers, FedBABU (Oh et al., 2021) proposed to finetune the classifiers and SphereFed (Dong et al., 2022) proposed a closed-form classifier calibration.

**FL with pretrained models.** Recently, there has been increasing interest in incorporating pre-trained, foundation models into federated learning. Multiple works (Nguyen et al., 2023; Tan et al., 2022b; Chen et al., 2022; Qu et al., 2022; Shysheya et al., 2022) propose using pre-trained weights which reduces the impact of client data heterogeneity and achieves faster model convergence. FedFN (Kim et al., 2023) recently combined feature normalization with FedAvg to enhance performance and highlighted that using pre-trained weights can sometimes negatively affect in highly heterogeneous settings. FedNCM (Legate et al., 2023a) share clients' class means for global classifier initialization without any training. Recently, Fed3R (Fanì et al., 2024) to initialize the global

classifier by using second-order feature statistics from all clients. Our work use pre-trained models and propose how we can estimate class covariances from only client means and exploit them.

# 3 PRELIMINARIES

Here, we introduce the general FL problem and the training-free approach via classifier initialization.

## 3.1 PROBLEM FORMULATION

In the FL setting, we assume $K$ clients have local datasets $D_k = (X_k, Y_k)$, where $k \in \{1, ..., K\}$. We denote the total number of images from all clients as $N$ where $N = \sum_{k=1}^{K} M_k$ and $M_k$ refers to the number of images in client $k$. We represent the model as $h_W(f_\theta(x))$ which can be decomposed into two parts: the feature extractor $f$ parameterized by $\theta$ which gives a $d$-dimensional embedding from a given image and the final classifier layer $h : \mathbb{R}^d \to \mathbb{R}^C$ parameterized by $W$ where $C$ refers to the total number of classes. The general objective of federated optimization (Konečnỳ et al., 2016) is to learn a global model that minimizes the sum of the losses across all the clients as follows:

$$\arg \min_{\theta, W} \sum_{k=1}^{K} \frac{M_k}{N} \mathcal{L}(h_W(f_\theta(X_k)), Y_k) \tag{1}$$

where $\mathcal{L}$ is the classification loss function (e.g., cross-entropy). With the growing quality of pre-trained models, recent works on FL (Chen et al., 2022; Legate et al., 2023a; Nguyen et al., 2023; Tan et al., 2022b; Fanì et al., 2024) has focused on scenarios where all clients start with a strong pre-trained network. After initializing $\theta$ with pre-trained weights, the models can be optimized in federated manner. In this work, we propose to use the frozen pre-trained model without performing local updates across clients.

## 3.2 TRAINING-FREE FEDERATED LEARNING METHODS

**Federated NCM.** Legate et al. (2023a) propose a Nearest Class Mean (NCM) classifier where the global classifier weights (considering a linear layer) for class $c$ denoted by $W_c$ can be initialized with normalized global class means $\hat{\mu}_c$, which are aggregated from the local class means $\hat{\mu}_{k,c}$ as follows:

$$W_c = \frac{\hat{\mu}_c}{\|\hat{\mu}_c\|}; \qquad \hat{\mu}_c = \frac{1}{N_c} \sum_{k=1}^{K} n_{k,c} \, \hat{\mu}_{k,c}; \qquad \hat{\mu}_{k,c} = \frac{1}{n_{k,c}} \sum_{x \in X_{k,c}} f(x) \tag{2}$$

where $X_{k,c}$ is the subset of $X_k$ containing only images of class $c$, $n_{k,c}$ refers to number of images in $X_{k,c}$ and $N_c = \sum_{k=1}^{K} n_{k,c}$ is the total number of images of class $c$ across all clients. FedNCM needs to communicate class means $\hat{\mu}_{k,c}$ and class counts $n_{k,c}$ from clients to the server only once.

**Federated Ridge Regression.** While FedNCM exploits only class means, Fed3R (Fanì et al., 2024) recently proposed to use ridge regression which needs second-order feature statistics from all clients to initialize the global classifier, leading to improved performance compared to FedNCM. The ridge regression problem aims to find the optimal weights that minimize the following objective:

$$W^* = \arg \min_{W \in \mathbb{R}^{d \times C}} \|Y - F^\top W\|^2 + \lambda \|W\|^2, \tag{3}$$

where $F \in \mathbb{R}^{d \times N}$ is the feature matrix extracted from a pre-trained model and $Y \in \mathbb{R}^{N \times C}$ contains one-hot encoding labels for the $N$ features with $C$ classes. The closed-form solution is given by:

$$W^* = (G + \lambda I_d)^{-1} B, \tag{4}$$

with $G = FF^\top \in \mathbb{R}^{d \times d}$ and $B = FY \in \mathbb{R}^{d \times C}$, $\lambda \in \mathbb{R}$ is an hyper-parameter and $I_d$ is the $d \times d$ identity matrix.

In Fed3R (Fanì et al., 2024), each client $k$ computes two local matrices $G_k = F_k F_k^\top \in \mathbb{R}^{d \times d}$ and $B_k = F_k Y_k \in \mathbb{R}^{d \times C}$, where $F_k$ and $Y_k$ are the feature matrix and the labels of client $k$, and then sends them to the global server. The server aggregates these matrices as

$$G = \sum_{k=1}^{K} G_k, \quad B = \sum_{k=1}^{K} B_k \tag{5}$$

and compute $W^*$ (Eq.4), which is normalized and then used to initialize the global linear classifier.

Table 1: FedNCM (Legate et al., 2023a) shares only class means $\hat{\mu}_{k,c}$ and has minimal communication. Fed3R (Fanì et al., 2024) requires sum of class features $B_k$ and feature matrix $G_k$ from all clients, thereby increasing the communication budget by $d^2K$. We propose FedCOF, which shares only class means and estimates a global class covariance $\hat{\Sigma}_c$ to initialize the classifier weights.

| Classifier Initialization | Each Client Shares | Server Uses | Communication Cost |
|---|---|---|---|
| FedNCM (Legate et al., 2023a) | $\{\hat{\mu}_{k,c}\}_{c=1}^{C}$ | $\{\{\hat{\mu}_{k,c}\}_{c=1}^{C}\}_{k=1}^{K}$ | $dCK$ |
| Fed3R (Fanì et al., 2024) | $G_k, B_k$ | $\{G_k, B_k\}_{k=1}^{K}$ | $(dC + d^2)K$ |
| FedCOF (ours) | $\{\hat{\mu}_{k,c}\}_{c=1}^{C}$ | $\{\{\hat{\mu}_{k,c}\}_{c=1}^{C}\}_{k=1}^{K}, \{\hat{\Sigma}_c\}_{c=1}^{C}$ | $dCK$ |

## 4 FEDERATED LEARNING WITH COVARIANCES FOR FREE (FEDCOF)

### 4.1 MOTIVATION

**Communication cost.** While Fed3R is more effective than FedNCM, it requires each client to send $C$ vectors of size $d$ and a $d \times d$ matrix, significantly increasing the communication overhead by $d^2K$ compared to FedNCM which only shares the class means (see Table 1). Scaling the method to a large number of clients requires much higher communication cost, as it scales linearly with number of clients and quadratically with the feature dimension, as shown in Fig. 2. Smaller neural network models often have a very high-dimensional feature space. For instance, ResNet-50 has $d = 2048$ with 25.6 million parameters, MobileNetV2 has $d = 1280$ with 3.4 million parameters while ViT-B/16 has more parameters (86 million) with $d = 768$. Considering cross-device FL settings (Kairouz et al., 2021), having millions of client devices, the communication cost needed for Fed3R would be enormous. In settings with low-bandwidth communication, using Fed3R is not realistic. See Appendix E for more discussion.

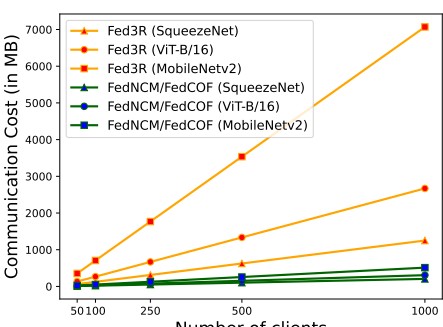

Figure 2: Analysis showing increasing communication cost for Fed3R with increasing number of clients assuming 100 classes per client. This is due to the high dimensionality of the features ($d = 512$ for SqueezeNet, $d = 768$ for ViT-B/16 and $d = 1280$ for MobileNetV2).

**Potential privacy concerns.** Sharing only class means provides a higher level of data privacy compared to sharing raw data, as prototypes represent the mean of feature representations. It is not easy to reconstruct exact images from prototypes with feature inversion attacks, as shown by (Luo et al., 2021). As a result, sharing class means is common in many recent works (Tan et al., 2022b;a; Shysheya et al., 2022; Legate et al., 2023a). However, sharing feature matrices (Fanì et al., 2024) exposes the feature distribution of the client to the server since all clients employ the same frozen pre-trained model to extract features. Sharing covariances makes the clients more vulnerable to attacks if additional secure aggregation protocols are not implemented (Bonawitz et al., 2016).

While exploiting second-order statistics (using Fed3R (Fanì et al., 2024), for example) yields significant gains in accuracy as shown in Fig. 1, it faces the above mentioned issues. We propose instead to estimate class covariances at the server using only class means and counts from clients. This will allow us to exploit second-order statistics without actually sharing them between clients and server. Following (Legate et al., 2023a; Luo et al., 2021), we use class frequencies from clients[1] since it only quantifies the client data while not revealing any information at the data or feature level.

### 4.2 ESTIMATING CLASS COVARIANCES USING ONLY CLIENT MEANS

We aim to use the pre-trained feature extractor and initialize the global classifier by exploiting second-order statistics for each class. We propose estimating the client class covariances using only class means. Our method leverages the statistical properties of sample means to derive an unbiased

---

[1]Our method requires communication of class frequencies which could raise minor privacy concerns; in Appendix L we perform an extensive evaluation of potential methods to address those concerns.

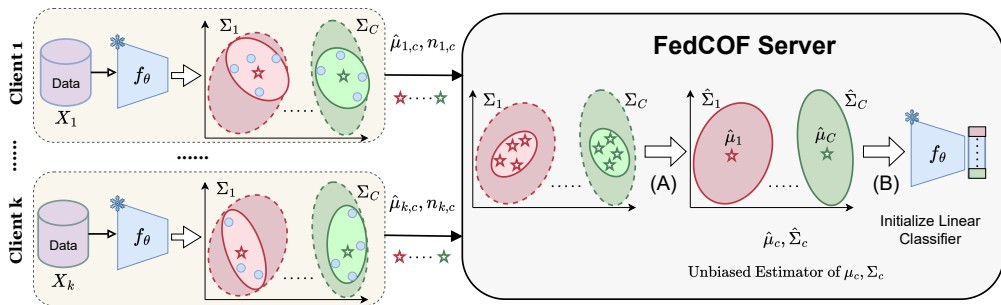

Figure 3: Federated Learning with COvariances for Free (FedCOF). Each client $k$ communicates only its class means $\hat{\mu}_{k,c}$ and counts $n_{k,c}$. On the server side, (A) we use a provably unbiased estimator $\hat{\Sigma}_c$ (denoted by solid lines) of population covariance $\Sigma_c$ (denoted by dashed lines) based on the received class means (see Section 4.2). (B) We initialize the linear classifier using the estimated second-order statistics and remove the between-class scatter matrix as discussed in Section 4.3.

estimator of the class population covariance based only on class means. This estimator is then used to initialize the classifier (see Fig. 3 for an overview).

Assume that the features of a class $c$ are drawn from a population with mean $\mu_c$ and covariance $\Sigma_c$. The features computed by each client are a random sample drawn from this population distribution. Using the statistical properties of the sample mean we can prove the following proposition.

**Proposition 1.** *Let* $\{F_{k,c}^j\}_{j=1}^{n_{k,c}}$ *be a random sample from a multivariate population with mean* $\mu_c$ *and covariance* $\Sigma_c$, *where* $F_{k,c}^j$ *is the j-th feature vector of class c assigned to the client k and* $n_{k,c}$ *is the number of elements of class c in the client k. Assuming that the per-class features* $F_{k,c}^j$ *in each client are iid in the initialization, then the sample mean of the features for class c*

$$\overline{F}_{k,c} = \frac{1}{n_{k,c}} \sum_{j=1}^{n_{k,c}} F_{k,c}^j, \tag{6}$$

*is distributed with mean* $\mathbb{E}[\overline{F}_{k,c}] = \mu_c$ *and covariance* $\mathrm{Var}[\overline{F}_{k,c}] = \frac{\Sigma_c}{n_{k,c}}$.

In Appendix B we provide the proof of this well-known result about the distribution of sample means and covariances. Intuitively, since $\Sigma_c = n_{k,c}\mathrm{Var}[\overline{F}_{k,c}]$, this proposition suggests that by assigning multiple sets of $n_{k,c}$ features to a single client, we can compute the empirical covariance of the client's class means over multiple assignments, providing an estimator of population covariance $\Sigma_c$.

However, in federated learning data are assigned only once to each client, and there are $K$ clients in the federation, each with $n_{k,c}$ features and $n_{i,c} \neq n_{j,c}$ for $i \neq j$. To estimate the population covariance $\Sigma_c$, we need an estimator that accounts for the contributions of all $K$ clients. In the following proposition, we propose such an estimator.

**Proposition 2.** *Let* $K$ *be the number of clients, each with* $n_{k,c}$ *features, and let* $C$ *be the total number of classes. Let* $\hat{\mu}_c = \frac{1}{N_c} \sum_{j=1}^{N_c} F^j$ *be the unbiased estimator of the population mean* $\mu_c$ *and* $N_c = \sum_{k=1}^{K} n_{k,c}$ *be the total number of features for a single class. Assuming the features for class c are iid across clients at initialization, the estimator*

$$\hat{\Sigma}_c = \frac{1}{K-1} \sum_{k=1}^{K} n_{k,c}(\overline{F}_{k,c} - \hat{\mu}_c)(\overline{F}_{k,c} - \hat{\mu}_c)^\top \tag{7}$$

*is an unbiased estimator of the population covariance* $\Sigma_c$, *for all* $c \in 1, \ldots, C$.

To prove that $\hat{\Sigma}_c$ is an unbiased estimator of the population covariance, we show in Appendix C that $\mathbb{E}[\hat{\Sigma}_c] = \Sigma_c$. Under the iid assumption of client feature distribution before federated training with a pre-trained model, the class features of each client can be considered as a random sample of size $n_{k,c}$, and the global class features as a sample of size $N_c$. By applying Proposition 1, we find that

each client class mean has $\mathbb{E}[\overline{F}_{k,c}] = \mu_c$ and $\text{Var}[\overline{F}_{k,c}] = \frac{\Sigma_c}{n_{k,c}}$, while the global class mean $\hat{\mu}_c$ has $\mathbb{E}[\hat{\mu}_c] = \mu_c$ and $\text{Var}[\hat{\mu}_c] = \frac{\Sigma_c}{N_c}$. Using the properties of expectation, we complete the proof.

**Covariance shrinkage.** Van Ness (1980) and Friedman (1989) proposed adding an identity matrix to the covariance matrix to stabilize the smaller eigenvalues. Shrinkage helps especially when the number of samples is fewer than the number of feature dimensions resulting in a low-rank covariance matrix. Shrinkage has been adopted in recent works on continual learning using feature covariances (Goswami et al., 2023; Magistri et al., 2024). In our problem, the proposed covariance estimation using a limited number of clients may poorly estimate the population covariance $\Sigma_c$. So, we perform shrinkage to better estimate the class covariances from the client means as follows:

$$\hat{\Sigma}_c = \frac{1}{K-1} \sum_{k=1}^{K} n_{k,c}(\hat{\mu}_{k,c} - \hat{\mu}_c)(\hat{\mu}_{k,c} - \hat{\mu}_c)^\top + \gamma I_d \tag{8}$$

where $\hat{\mu}_{k,c} = \overline{F}_{k,c}$ represents a realization of client means and $\gamma > 0$ is the shrinkage factor.

**Impact of the number of clients.** The quality of estimated covariances depends on number of clients. More clients will give more class means, thus improving the estimate compared to fewer clients. While realistic FL settings has thousands of clients (Hsu et al., 2020; Kairouz et al., 2021), there can be FL settings with fewer clients. In cases with few clients, we propose to sample multiple means from each client to increase total number of means used for covariance estimation. This can be easily done by randomly sampling multiple subsets of features in each client without replacement and computing a mean from each of these subsets. We validate this in experiments (see Fig. 8a).

**The iid assumption.** In FL each client has its own data, typically distributed in a statistically heterogeneous or class-imbalanced manner according to a Dirichlet distribution (Hsu et al., 2019). As a result, each client has data belonging to a different set of classes in varying quantities, resulting in non-iid data distributions across clients. However, note that the samples belonging to the same class in different clients are sampled from the same distribution. We exploit this fact in FedCOF. We later show empirically that our method can be successfully applied to non-iid FL scenarios involving thousands of heterogeneous clients on iNaturalist-Users-120K (Hsu et al., 2020). We hypothesize that this can be attributed to the strong generalization capabilities of pre-trained models, which help in moving the distribution of class samples across clients towards an iid feature distribution even if the class distribution across clients is non-iid (Nguyen et al., 2023). We analyze the bias of the estimator under non-iid assumptions in Appendix J and evaluate the performance of FedCOF in feature shift settings (Li et al., 2021) in Appendix K.

### 4.3 CLASSIFIER INITIALIZATION WITH ESTIMATED COVARIANCES

Having derived how to compute class covariances from client means, we now discuss how to use class covariances to set the classifier weights and then replace the empirical class covariances with our estimated class covariances, as illustrated in Fig. 3.

**Proposition 3.** *Let $F \in \mathbb{R}^{d \times N}$ be a feature matrix with empirical global mean $\hat{\mu}_g \in \mathbb{R}^d$, and $Y \in \mathbb{R}^{N \times C}$ be a label matrix. The optimal ridge regression solution $W^* = (G + \lambda I_d)^{-1}B$, where $B \in \mathbb{R}^{d \times C}$ and $G \in \mathbb{R}^{d \times d}$ can be written in terms of class means and covariances as follows:*

$$B = [\hat{\mu}_c N_c]_{c=1}^C, \tag{9}$$

$$G = \sum_{c=1}^{C} (N_c - 1)\hat{S}_c + \sum_{c=1}^{C} N_c(\hat{\mu}_c - \hat{\mu}_g)(\hat{\mu}_c - \hat{\mu}_g)^\top + N\hat{\mu}_g\hat{\mu}_g^\top \tag{10}$$

*where the first two terms $\sum_{c=1}^{C}(N_c - 1)\hat{S}_c$ and $\sum_{c=1}^{C} N_c(\hat{\mu}_c - \hat{\mu}_g)(\hat{\mu}_c - \hat{\mu}_g)^\top$ represents the within-class and between class scatter respectively, while $\hat{\mu}_c$, $\hat{S}_c$ and $N_c$, denote the empirical mean, covariance and sample size for class c, respectively.*

We prove this result in Appendix D. The proof is based on the key observation that $G = FF^\top$ from ridge regression is an uncentered and unnormalized empirical global covariance. By employing the empirical global covariance definition and decomposing it into within-class and between-class scatter matrices, we obtain the above formulation of $G$.

---

**Algorithm 1** Federated Learning with COvariances for Free (FedCOF)

| **Client-Side** (Client $k$): | **Server-Side:** |
|---|---|
| **Input:** | **Input:** |
| $C$: set of all classes | $\hat{\mu}_{k,c}, n_{k,c}$ sent from $K$ clients |
| $f_\theta$: pre-trained feature extractor | $\lambda > 0, \gamma > 0$: hyper-parameters |
| $n_{k,c}$: number of samples for class $c$ in | **for** $c = 1 \dots C$ **do** |
| client $k$ | $\hat{\mu}_c = \frac{1}{N_c} \sum_{k=1}^{K} n_{k,c} \hat{\mu}_{k,c}; N_c = \sum_{k=1}^{K} n_{k,c}$ # class mean |
| $X_{k,c}$: samples of class $c$ in client $k$ | |
| **for** $c = 1$ to $C$ **do** | $\hat{\Sigma}_c = \frac{1}{K-1} \sum_{k=1}^{K} n_{k,c}(\hat{\mu}_{k,c} - \hat{\mu}_c)(\hat{\mu}_{k,c} - \hat{\mu}_c)^\top + \gamma I_d$, Eq.(8) |
| $\quad \hat{\mu}_{k,c} = \frac{1}{n_{k,c}} \sum_{x \in X_{k,c}} f_\theta(x)$ | **end for** |
| **end for** | $\hat{\mu}_g = \frac{1}{N} \sum_{c=1}^{C} N_c \hat{\mu}_c \qquad N = \sum_{c=1}^{C} N_c$ # global mean |
| **Send** the class means $\hat{\mu}_{k,c}$ and sample | $B = [\hat{\mu}_c N_c]_{c=1}^{C}$, Eq.(9) |
| counts $n_{k,c}$ to the Server | $\hat{G} = \sum_{c=1}^{C} (N_c - 1)\hat{\Sigma}_c + N\hat{\mu}_g \hat{\mu}_g^\top$ |
| | $W^* = (\hat{G} + \lambda I_d)^{-1} B$, Eq. (11) |
| | **Normalize** $W^*$: $W_c^* \leftarrow W_c^*/\|W_c^*\| \quad c = 1, \dots, C$ |

---

To analyze the impact of the two scatter matrices, we consider the centralized setting in Fig. 4 and empirically find that using only within-class scatter matrix performs slightly better than using total scatter matrix in Eq. (10). As a result, we propose to remove the between-class scatter and initialize the linear classifier at the end of the pre-trained network using the within-class covariances $\hat{\Sigma}_c$ which are estimated from client means using Eq. (8), as follows:

$$W^* = (\hat{G} + \lambda I_d)^{-1}B; \qquad \hat{G} = \sum_{c=1}^{C} (N_c - 1)\hat{\Sigma}_c + N\hat{\mu}_g \hat{\mu}_g^\top. \tag{11}$$

Theoretically, we observe that a similar approach is used in Linear Discriminant Analysis (Ghojogh & Crowley, 2019), which employs only within-class covariances for finding optimal weights.

To summarize, we estimate the covariance matrix for each class using only the client means (Eq. (8)) and use the estimated covariances to initialize the classifier as in Eq. (11). Finally, we normalize the weights for every class to account for class imbalance in the entire dataset. We provide the summary in Algorithm 1.

**FedCOF with multiple rounds.** While the proposed estimator requires class means from all clients in a single round, this might not be realistic in settings in which clients appear in successive rounds based on availability. In the case of multi-round classifier initialization (see FedCOF in Fig. 6 before fine-tuning), the server uses all class means and counts received from all clients seen up to the current round and stores the accumulated means and counts for future use. As a result, FedCOF uses statistics from all clients seen up to the current round, similar to Fed3R. Thus FedCOF converges when all clients are seen at least once.

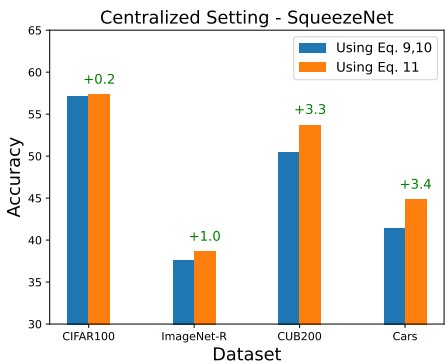

Figure 4: Analysis showing improved accuracy by removing between-class scatter for classifier weights using Eq. (11) in centralized setting.

## 5 EXPERIMENTS

**Datasets.** We evaluate the proposed method on multiple datasets namely CIFAR-100 (Krizhevsky, 2009), ImageNet-R (Hendrycks et al., 2021), CUB200 (Wah et al., 2011), Stanford Cars (Krause et al., 2013) and iNaturalist (Van Horn et al., 2018). CIFAR-100 has 100 classes provided in 50k training and 10k testing images. ImageNet-R (IN-R) is composed of 30k images covering 200 ImageNet classes. CUB200 has as well 200 classes of different bird species provided in 5994 training and 5794 testing images. Stanford Cars has 196 classes of cars with 8144 training images and 8041 test images. We distribute these datasets to 100 clients using a highly heterogeneous Dirichlet distribution ($\alpha = 0.1$) following standard practice (Hsu et al., 2019; Legate et al., 2023a). We also

Table 2: Evaluation of different training-free methods using 100 clients for four datasets and 9275 pre-defined clients on iNat-120K using 5 random seeds. We show the total communication cost (in MB) from all clients to server. We also show the FedCOF oracle in which full class covariances are shared from clients to server. The best results from each section are highlighted in **bold**.

| | Method | SqueezeNet ($d = 512$) | | MobileNetv2 ($d = 1280$) | | ViT-B/16 ($d = 768$) | |
|---|---|---|---|---|---|---|---|
| | | Acc (↑) | Comm. (↓) | Acc (↑) | Comm. (↓) | Acc (↑) | Comm. (↓) |
| **CIFAR100** | FedNCM (Legate et al., 2023a) | 41.5±0.1 | **5.9** | 55.6±0.1 | **14.8** | 55.2±0.1 | **8.9** |
| | Fed3R (Fanì et al., 2024) | **56.9**±0.1 | 110.2 | 62.7±0.1 | 670.1 | **73.9**±0.1 | 244.8 |
| | FedCOF (Ours) | 56.1±0.2 | **5.9** | **63.5**±0.1 | **14.8** | 73.2±0.1 | **8.9** |
| | FedCOF Oracle (Full Covs) | 56.4±0.1 | 3015.3 | 63.9±0.1 | 18823.5 | 73.8±0.1 | 6780.0 |
| **IN-R** | FedNCM (Legate et al., 2023a) | 23.8±0.1 | **7.1** | 37.6±0.2 | **17.8** | 32.3±0.1 | **10.7** |
| | Fed3R (Fanì et al., 2024) | 37.6±0.2 | 111.9 | 46.0±0.3 | 673.1 | **51.9**±0.2 | 246.6 |
| | FedCOF (Ours) | **37.8**±0.4 | **7.1** | **47.4**±0.1 | **17.8** | 51.8±0.3 | **10.7** |
| | FedCOF Oracle (Full Covs) | 38.2±0.1 | 3645.7 | 48.0±0.3 | 22758.8 | 52.7±0.1 | 8197.4 |
| **CUB200** | FedNCM (Legate et al., 2023a) | 37.8±0.3 | **4.8** | 58.3±0.3 | **12.0** | 75.7±0.1 | **7.2** |
| | Fed3R (Fanì et al., 2024) | 50.4±0.3 | 109.6 | 58.6±0.2 | 667.3 | 77.7±0.1 | 243.1 |
| | FedCOF (Ours) | **53.7**±0.3 | **4.8** | **62.5**±0.4 | **12.0** | **79.4**±0.2 | **7.2** |
| | FedCOF Oracle (Full Covs) | 54.4±0.1 | 2472.1 | 63.1±0.5 | 15432.7 | 79.6±0.2 | 5558.6 |
| **Cars** | FedNCM (Legate et al., 2023a) | 19.8±0.2 | **5.4** | 30.0±0.1 | **13.5** | 26.2±0.4 | **8.1** |
| | Fed3R (Fanì et al., 2024) | 39.9±0.2 | 110.2 | 41.6±0.1 | 668.8 | 47.9±0.3 | 244.0 |
| | FedCOF (Ours) | **44.0**±0.3 | **5.4** | **47.3**±0.5 | **13.5** | **52.5**±0.3 | **8.1** |
| | FedCOF Oracle (Full Covs) | 44.6±0.1 | 2767.3 | 47.2±0.3 | 17275.7 | 53.1±0.1 | 6222.5 |
| **iNat-120K** | FedNCM (Legate et al., 2023a) | 21.2±0.1 | **111.8** | 36.0±0.1 | **279.5** | 53.9±0.1 | **167.7** |
| | Fed3R (Fanì et al., 2024) | 32.1±0.1 | 9837.3 | 41.5±0.1 | 61064.1 | 62.5±0.1 | 22050.2 |
| | FedCOF (Ours) | **32.5**±0.1 | **111.8** | **44.1**±0.1 | **279.5** | **63.1**±0.1 | **167.7** |
| | FedCOF Oracle (Full Covs) | 32.4±0.1 | 57k | 43.6±0.1 | 358k | 62.9±0.1 | 128k |

use the real-world non-iid FL benchmark of iNaturalist-Users-120K (Hsu et al., 2020) (iNat-120K) having 1203 classes across 9275 clients and 120k training images.

**Implementation Details.** We use three models: namely SqueezeNet (Iandola et al., 2016) following Legate et al. (2023a) and Nguyen et al. (2023), MobileNetV2 (Sandler et al., 2018) following Fanì et al. (2024); Hsu et al. (2020), and ViT-B/16 (Dosovitskiy et al., 2021). All models are pre-trained on ImageNet-1k (Deng et al., 2009). We use the FLSim library for our experiments and implement all methods in the same setting. We use $\gamma = 1$ for all experiments with SqueezeNet and ViT-B/16, and $\gamma = 0.1$ for all experiments with MobileNetV2 due to very high dimensionality $d$ of the feature space. Following Fanì et al. (2024), we use $\lambda = 0.01$ for both Fed3R and FedCOF. We also include the oracle setting of FedCOF in which the full class covariances are shared from clients to server and aggregated similar to Luo et al. (2021). We discuss this oracle further in Appendix G. We also evaluate how FedCOF impact further fine-tuning (FT) or linear probing (LP) of the models. For all FT and LP experiments, we train 100 client models for 200 rounds, 1 local epochs per round, and set the client participation to 30%. We provide more details in Appendix F. We will release the code after the review phase.

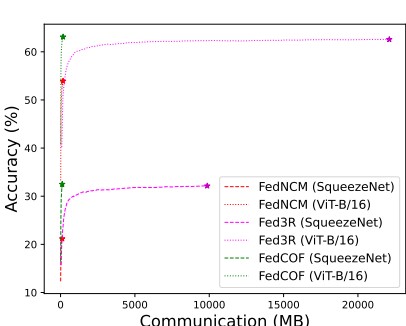

Figure 5: Performance vs. communication cost analysis using pre-trained SqueezeNet and ViT-B/16 on iNaturalist-Users-120K.

## 5.1 EVALUATION FOR DIFFERENT TRAINING-FREE METHODS

We compare the performance of existing training-free methods and the proposed method in Table 2 using pre-trained Squeezenet, Mobilenetv2 and ViT-B/16 models. We observe that Fed3R (Fanì et al., 2024) using second-order statistics outperforms FedNCM (Legate et al., 2023a) by significant margins ranging from 0.3% to 21% across all datasets. However, Fed3R requires a higher communication cost compared to FedNCM. In real-world iNat-120K benchmark, Fed3R needs $61k$ MB compared to 280 MB for FedNCM (see Fig. 5), which is 218 times higher. FedCOF performs better

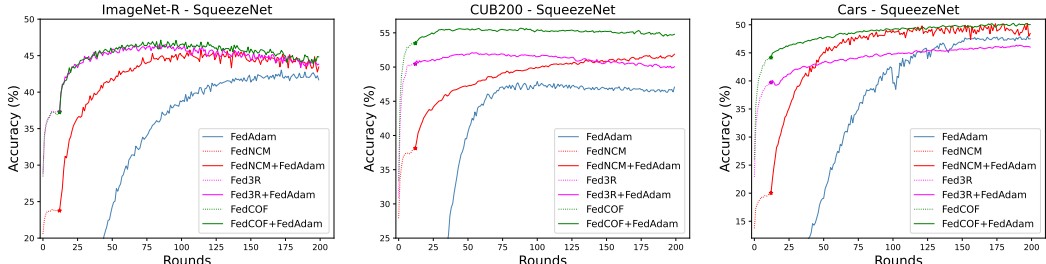

Figure 6: Analysis of performance when initialized with different methods and then fine-tuned with FedAdam (Reddi et al., 2020) using a pre-trained SqueezeNet model. Here, FedAdam (in blue) uses random classifier initilization and pre-trained backbone.

than Fed3R in most settings despite having the same communication cost as FedNCM. FedCOF achieves very similar performance as the oracle setting using aggregated class covariances requiring very high communication, which validates the effectiveness of the proposed method. Note that in Table 2 the number of means used to estimate each class covariance is less than the total number of clients, e.g. approximately 11.8 means per class are used for estimation in CUB200. This is due to the fact that not all classes are present in all clients.

FedCOF maintains similar accuracy with Fed3R on CIFAR100 and ImageNet-R, with an improvement of about 1% when using MobileNetv2. FedCOF performs significantly better than Fed3R on CUB200 and Cars. On CUB200, FedCOF outperforms Fed3R by 3.3%, 3.9% and 2.2% using SqueezeNet, MobileNetv2 and ViT-B/16 respectively. FedCOF improves over Fed3R in the range of 4.1% to 5.7% on Cars. On iNat-120K, FedCOF improves over Fed3R by 0.4%, 2.6% and 0.6% using different models, and even performs marginally better than the oracle setting which can be attributed to less noisy covariance estimation from large number of client means. When comparing FedCOF with FedNCM – both with equal communication costs and same strategy in clients – one can observe that the usage of second order statistics derived only from the class means of clients leads to large performance gains, e.g. 24% using SqueezeNet and 26% using ViT-B/16 on Cars, about 10% using all architectures on large-scale iNat-120K.

## 5.2 ANALYSIS OF FINE-TUNING AND LINEAR PROBING

While we achieve very high accuracy without any training with FedCOF, we show in Fig. 6 that further fine-tuning the model with FL optimization methods achieves better and faster convergence compared to federated optimization from scratch. We use adaptive optimizer, FedAdam (Reddi et al., 2020) for all FT experiments since it performs better than other optimizers as discussed in (Nguyen et al., 2023). We show in Fig. 6, how the performance of FedNCM, Fed3R, and the proposed FedCOF evolves over multiple rounds. These training-free methods end after all clients appear at least once to share their local statistics to server. We observe that fine-tuning after performing classifier initialization with FedCOF starts with a higher accuracy and converges faster and better compared to FedNCM. While FedCOF and Fed3R initial-

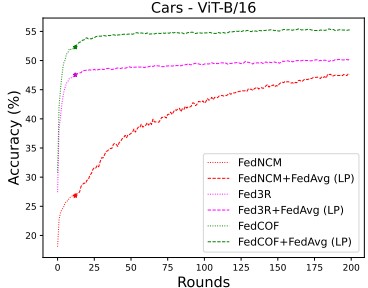

Figure 7: Linear probing with FedAvg (McMahan et al., 2017) using pre-trained ViT-B/16.

ization converges similarly on ImageNet-R, FedCOF+FedAdam achieves a better accuracy than Fed3R+FedAdam on CUB200 and Cars. All the training-free classifier initialization approaches outperform FedAdam with a random classifier initialization and pre-trained backbone.

Following (Legate et al., 2023a; Nguyen et al., 2023), we also perform federated linear probing of the models using FedAvg (McMahan et al., 2017) after classifier initialization with different training-free methods. Linear probing requires much less computation compared to fine-tuning the entire network and were found to be effective with pre-trained models. We observe that linear probing after FedCOF improves significantly compared to FedNCM and Fed3R using ViT-B/16 Fig. 7 on Cars. We provide more experiments in Appendix H. The improved performance with FedCOF initialization for finetuning and probing validates the effectiveness of the proposed method.

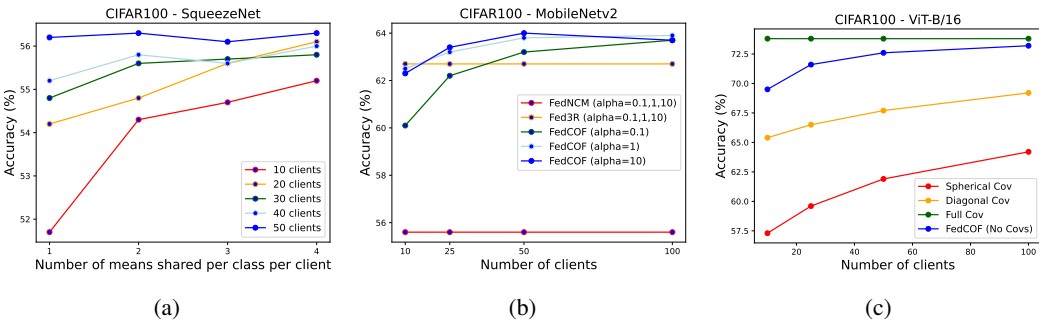

(a)                                      (b)                                      (c)

Figure 8: Ablation experiments on CIFAR-100: (a) Shows that communicating multiple class means per client improves FedCOF performance with fewer clients. (b) Shows how the performance of the proposed method changes with the number of clients and varying data heterogeneity. (c) Performance comparison of FedCOF with full, diagonal, and spherical covariance matrix communication.

## 5.3 ABLATION STUDIES

**Communicating multiple class means per client.** We analyze the FL settings with fewer clients ranging from 10 to 50 clients in Fig. 8a and show that sharing more than 1 class means from each client by subsampling data and computing multiple means in each client improves the performance. Using only 10 clients, sharing 2 class means per client improves the accuracy by 2.6%.

**Impact of the number of clients and data heterogeneity.** We analyze the impact of the number of clients and data heterogeneity on the performance of FedCOF using MobileNetv2 on CIFAR-100 as shown in Fig. 8b. We observe that the performance of FedCOF improves with increasing number of clients and decreasing heterogeneity. This is due to the fact that more clients provides more class means and more uniform data distribution gives better representative local means. While more number of clients are favourable for FedCOF, it still performs well and outperforms FedNCM significantly in the setting with only 10 clients and high data heterogeneity.

**Communicating diagonal or spherical covariances.** While communicating diagonal or spherical covariances (mean of the diagonal covariance) from clients to server and then estimating the global class covariance from them can significantly reduce the communication cost, such estimates of global class covariance is poor compared to FedCOF. We show in Fig. 8c that FedCOF outperforms these covariance sharing baselines when communicating either spherical or diagonal covariances on CIFAR-100 using ViT-B/16.

## 6 CONCLUSION

In this work we proposed FedCOF, a novel training-free approach for federated learning with pretrained models. By leveraging the statistical properties of distribution client class sample means, we show that second-order statistics can be estimated using only class means from clients, thus reducing communication costs. We derive a provably unbiased estimator of population class covariances, enabling accurate estimation of a global covariance matrix. After applying shrinkage to the estimated class covariances and removing between-class scatter matrices, we show that the server can effectively use this global covariance to initialize the global classifier. Our experimental results show that FedCOF outperforms FedNCM (Legate et al., 2023a) by significant margins while maintaining the same communication costs. Additionally, FedCOF delivers competitive or even superior results to Fed3R (Fanì et al., 2024) across various model architectures and benchmarks while substantially reducing communication costs. Moreover, we empirically demonstrate that our approach can serve as a more effective starting point for improving the convergence of standard federated fine-tuning and linear probing methods.

**Limitations.** The quality of our estimator depends on the number of clients, as shown in Fig. 8a where using multiple class means per client helps with fewer client settings. Another limitation is the assumption that samples of the same class are iid across clients, which is however an assumption underlying most of federated learning. Finally, our method assumes the existence of a pre-trained network. If the domain shift with the client data is sufficiently large, this is expected to impact the performance.

REPRODUCIBILITY STATEMENT

We have taken steps to ensure the reproducibility of our work. The full source code, along with scripts to reproduce all results in the paper, will be published after the review period. All experiments were performed on publicly available datasets, and details of model architectures, and main training hyperparameters are given in the main paper with additional details included in the supplementary material. To ensure the reproducibility of stochastic processes, such as weight initialization and dataset shuffling, we fix random seeds across all experiments. The random seed values will be clearly documented in our published code.

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

## APPENDIX A  SCOPE AND SUMMARY OF NOTATION

These appendices provide additional information, proofs, experimental results, and analyses that complement the main paper. For clarity and convenience, here we first summarize the key notations used throughout the paper:

- $N$: total number of samples.
- $K$: number of clients.
- $C$: number of classes.
- $d$: dimensionality of the feature space.
- $n_{k,c}$: number of samples from class $c$ assigned to client $k$.
- $N_c = \sum_{k=1}^{K} n_{k,c}$: total number of samples in class $c$.
- $\hat{\mu}_g, \hat{\mu}_c \in \mathbb{R}^d$: *empirical* global mean and class mean for class $c$, respectively.
- $\mu_c \in \mathbb{R}^d$: *population* mean of class $c$.
- $\hat{S}_c \in \mathbb{R}^{d \times d}$: *empirical* sample covariance for class $c$.
- $\Sigma_c \in \mathbb{R}^{d \times d}$: *population* covariance for class $c$.
- $\hat{\Sigma}_c \in \mathbb{R}^{d \times d}$: our unbiased estimator of the population covariance $\Sigma_c$ employing only client means.
- $F \in \mathbb{R}^{d \times N}$: feature matrix, where each column $F^j \in \mathbb{R}^d$ is a feature vector, for $j = 1, \dots, N$.
- $F_{k,c}^j \in \mathbb{R}^d$: $j$-th feature vector from class $c$ assigned to client $k$.
- $\overline{F}_{k,c} \in \mathbb{R}^d$: sample mean of the feature vectors for class $c$ on client $k$, treated as a random vector. A specific realization of this random vector is denoted by $\hat{\mu}_{k,c}$.
- $\mathrm{Var}[\overline{F}_{k,c}] = \mathrm{Cov}[\overline{F}_{k,c}, \overline{F}_{k,c}]$ represents the covariance matrix of the random vector $\overline{F}_{k,c}$.

## APPENDIX B  PROOF OF PROPOSITION 1

**Proposition 1.** *Let $\{F_{k,c}^j\}_{j=1}^{n_{k,c}}$ be a random sample from a multivariate population with mean $\mu_c$ and covariance $\Sigma_c$, where $F_{k,c}^j$ is the $j$-th feature vector of class $c$ assigned to the client $k$ and $n_{k,c}$ is the number of elements of class $c$ in the client $k$. Assuming that the per-class features $F_{k,c}^j$ in each client are iid in the initialization, then the sample mean of the features for class $c$*

$$\overline{F}_{k,c} = \frac{1}{n_{k,c}} \sum_{j=1}^{n_{k,c}} F_{k,c}^j, \tag{12}$$

*is distributed with mean $\mathbb{E}[\overline{F}_{k,c}] = \mu_c$ and covariance $\mathrm{Var}[\overline{F}_{k,c}] = \frac{\Sigma_c}{n_{k,c}}$.*

*Proof.* To prove this, we fix the class $c$ and omit the dependencies on $c$ for simplicity. Thus, we write $n_{k,c} = n_k$, $F_{k,c}^j = F_k^j$, $\overline{F}_{k,c} = \overline{F}_k$, and $\mu_c = \mu$, $\Sigma_c = \Sigma$.

Since $\{F_k^j\}_{j=1}^{n_k}$ is a random sample from a multivariate distribution with mean $\mu$ and covariance $\Sigma$, and the per-class features $F_k^j$ in each client are i.i.d at initialization, it follows that:

$$\mathbb{E}[F_k^j] = \mu \qquad \mathrm{Var}[F_k^j] = \Sigma, \quad \forall j \tag{13}$$

By computing the expectation of $\overline{F}_k$ and using the linearity of expectation, we obtain:

$$\mathbb{E}[\overline{F}_k] = \mathbb{E}[\frac{1}{n_k} \sum_{j=1}^{n_k} F_k^j] = \frac{1}{n_k} \mathbb{E}[F_k^1] + \dots + \frac{1}{n_k} \mathbb{E}[F_k^{n_k}] = \frac{1}{n_k}(n_k \mu) = \mu,$$

where in the last equality we used Eq. (13). Thus the expectation of the sample mean is $\mu$, which completes the first part of the proof.

Next, we show that the variance of the sample mean is $\frac{\Sigma}{n_k}$. By computing the variance of $\overline{F}_k$ and using the fact that the variance scales by the square of the constant, we obtain:

$$\text{Var}[\overline{F}_k] = \text{Var}[\frac{1}{n_k}\sum_{j=1}^{n_k} F_k^j] = \frac{1}{n_k^2}\left(\text{Var}[F_k^1] + \ldots + \text{Var}[F_k^{n_k}]\right) + \frac{1}{n_k^2}\sum_{i=1}^{n_k}\sum_{\substack{j=1 \\ j \neq i}}^{n_k} \text{Cov}[F_k^i, F_k^j].$$

By the independence assumption of $\{F_k^j\}_{j=1}^{n_k}$, the cross terms $\text{Cov}[F_k^i, F_k^j] = 0$ for $i \neq j$. Applying Equation 13, we have:

$$\text{Var}[\overline{F}_k] = \frac{1}{n_k^2}\left(\text{Var}[F_k^1] + \ldots + \text{Var}[F_k^{n_k}]\right) = \frac{1}{n_k^2}\left(n_k\Sigma\right) = \frac{\Sigma}{n_k}$$

$\square$

## APPENDIX C   PROOF OF PROPOSITION 2

**Proposition 2.** *Let $K$ be the number of clients, each with $n_{k,c}$ features, and let $C$ be the total number of classes. Let $\hat{\mu}_c = \frac{1}{N_c}\sum_{j=1}^{N_c} F^j$ be the unbiased estimator of the population mean $\mu_c$ and $N_c = \sum_{k=1}^{K} n_{k,c}$ be the total number of features for a single class. Assuming the features for class $c$ are iid across clients at initialization, the estimator*

$$\hat{\Sigma}_c = \frac{1}{K-1}\sum_{k=1}^{K} n_{k,c}(\overline{F}_{k,c} - \hat{\mu}_c)(\overline{F}_{k,c} - \hat{\mu}_c)^\top \tag{14}$$

*is an unbiased estimator of the population covariance $\Sigma_c$, for all $c \in 1, \ldots, C$.*

*Proof.* To prove this, we fix the class $c$ and omit the dependencies on $c$ for clarity. So we write $n_{k,c} = n_k, \overline{F}_{k,c} = \overline{F}_k, N_c = N, \hat{\mu}_c = \hat{\mu}, \hat{\Sigma}_c = \hat{\Sigma}, \mu_c = \mu$, and $\Sigma_c = \Sigma$. By the definition of an unbiased estimator, we need to show that:

$$\mathbb{E}[\hat{\Sigma}] = \mathbb{E}\left[\frac{1}{K-1}\sum_{k=1}^{K} n_k(\overline{F}_k - \hat{\mu})(\overline{F}_k - \hat{\mu})^\top\right] = \Sigma.$$

By the linearity of the expectation, the definition of sample mean $\overline{F}_k = \frac{1}{n_k}\sum_{j=1}^{n_k} F_k^j$, and the definition of global class mean $\hat{\mu} = \frac{1}{N}\sum_{k=1}^{K}\sum_{j=1}^{n_k} F_k^j$, we have:

$$\mathbb{E}[\hat{\Sigma}] = \frac{1}{K-1}\left(\sum_{k=1}^{K} n_k\mathbb{E}[\overline{F}_k\overline{F}_k^\top] - \sum_{k=1}^{K} n_k\mathbb{E}[\overline{F}_k\hat{\mu}^\top] - \sum_{k=1}^{K} n_k\mathbb{E}[\hat{\mu}\overline{F}_k^\top] + \sum_{k=1}^{K} n_k\mathbb{E}[\hat{\mu}\hat{\mu}^\top]\right)$$

$$= \frac{1}{K-1}\left(\sum_{k=1}^{K} n_k\mathbb{E}[\overline{F}_k\overline{F}_k^\top] - 2\mathbb{E}[(\sum_{k=1}^{K}\sum_{j=1}^{n_k} F_k^j)\hat{\mu}^\top] + \sum_{k=1}^{K} n_k\mathbb{E}[\hat{\mu}\hat{\mu}^\top]\right)$$

$$= \frac{1}{K-1}\left(\sum_{k=1}^{K} n_k\mathbb{E}[\overline{F}_k\overline{F}_k^\top] - 2N\mathbb{E}[\hat{\mu}\hat{\mu}^\top] + \sum_{k=1}^{K} n_k\mathbb{E}[\hat{\mu}\hat{\mu}^\top]\right). \tag{15}$$

By applying the variance definition and proposition 1, we obtain:

$$\mathbb{E}[\overline{F}_k\overline{F}_k^\top] = \text{Var}[\overline{F}_k] + \mathbb{E}[\overline{F}_k]\mathbb{E}[\overline{F}_k]^\top = \frac{\Sigma}{n_k} + \mu\mu^\top. \tag{16}$$

Now, by considering the right term in Eq. 15, since $\hat{\mu}$ is an unbiased estimator of the population mean, then $\mathbb{E}[\hat{\mu}] = \mu$. Moreover, since we assume that the features for a single class across clients are i.i.d at initialization, we can re-use Proposition 1 by considering the all class features as a random

sample of size $N$ from a population with mean $\mu$ and variance $\Sigma$. Consequently, the global sample mean $\hat{\mu}$ is has variance $\text{Var}[\hat{\mu}] = \frac{\Sigma}{N}$. Then

$$\mathbb{E}[\hat{\mu}\hat{\mu}^\top] = \text{Var}[\hat{\mu}] + \mathbb{E}[\hat{\mu}]\mathbb{E}[\hat{\mu}]^\top = \frac{\Sigma}{N} + \mu\mu^\top. \tag{17}$$

By replacing equations 16 and 17 in Eq. 15, and recalling that $N = \sum_{k=1}^K n_k$, we obtain:

$$\mathbb{E}[\hat{\Sigma}] = \frac{1}{K-1}\left(\sum_{k=1}^K n_k(\frac{\Sigma}{n_k} + \mu\mu^\top) - 2N(\frac{\Sigma}{N} + \mu\mu^\top) + \sum_{k=1}^K n_k(\frac{\Sigma}{N} + \mu\mu^\top)\right)$$

$$= \frac{1}{K-1}(K\Sigma + \mu\mu^\top N - 2\Sigma - 2N\mu\mu^\top + (\frac{\Sigma}{N} + \mu\mu^\top)N) = \frac{1}{K-1}(K-1)\Sigma = \Sigma.$$

$\square$

## APPENDIX D  PROOF OF PROPOSITION 3

**Proposition 3.** *Let $F \in \mathbb{R}^{d \times N}$ be a feature matrix with empirical global mean $\hat{\mu}_g \in \mathbb{R}^d$, and $Y \in \mathbb{R}^{N \times C}$ be a label matrix. The optimal ridge regression solution $W^* = (G + \lambda I_d)^{-1}B$, where $B \in \mathbb{R}^{d \times C}$ and $G \in \mathbb{R}^{d \times d}$ can be written in terms of class means and covariances as follows:*

$$B = [\hat{\mu}_c N_c]_{c=1}^C\,, \tag{18}$$

$$G = \sum_{c=1}^C (N_c - 1)\hat{S}_c + \sum_{c=1}^C N_c(\hat{\mu}_c - \hat{\mu}_g)(\hat{\mu}_c - \hat{\mu}_g)^\top + N\hat{\mu}_g\hat{\mu}_g^\top \tag{19}$$

*where the first two terms $\sum_{c=1}^C (N_c - 1)\hat{S}_c$ and $\sum_{c=1}^C N_c(\hat{\mu}_c - \hat{\mu}_g)(\hat{\mu}_c - \hat{\mu}_g)^\top$ represents the within-class and between class scatter respectively, while $\hat{\mu}_c$, $\hat{S}_c$ and $N_c$, denote the empirical mean, covariance and sample size for class c, respectively.*

*Proof.* The first part, regarding Eq. (18), follows directly. From the ridge regression solution, $B = FY$, which is obtained by summing the features for each class and arranging them into the columns of a matrix. This results in the product of class means and samples per class.

Now, for computing the matrix $G$, we proceed with the definition of the global sample covariance:

$$\hat{S} = \frac{1}{N-1}(F - \overline{F})(F - \overline{F})^\top = \frac{1}{N-1}\left(FF^\top - F\overline{F}^\top - \overline{F}F^\top + \overline{F}\,\overline{F}^\top\right),$$

where $\overline{F} = \left(\frac{1}{N}\sum_{j=1}^N F^j\right)\mathbf{1}^\top = \hat{\mu}_g\mathbf{1}^\top \in \mathbb{R}^{d \times N}$ is the matrix obtained by replicating the global mean $N$ times in each column and $\mathbf{1} \in \mathbb{R}^{N \times 1}$ is a column vector of ones. Recalling that $G = FF^\top$, we have:

$$\hat{S} = \frac{1}{N-1}(G - F\mathbf{1}\hat{\mu}_g^\top - \hat{\mu}_g\mathbf{1}^\top F^\top + \hat{\mu}_g\mathbf{1}^\top\mathbf{1}\hat{\mu}_g^\top) = \frac{1}{N-1}(G - 2F\mathbf{1}\hat{\mu}_g^\top + N\hat{\mu}_g\hat{\mu}_g^\top)$$

since $F\mathbf{1}\hat{\mu}_g^\top = \hat{\mu}_g\mathbf{1}^T F^\top$ and $\mathbf{1}^T\mathbf{1} = N$.

Now, since $F\mathbf{1} = \sum_{j=1}^N F^j$, we can obtain the matrix $G$ as:

$$G = (N-1)\hat{S} + 2\left(\sum_{j=1}^N F^j\right)\hat{\mu}_g^\top - N\hat{\mu}_g\hat{\mu}_g^\top = (N-1)\hat{S} + 2N\hat{\mu}_g\hat{\mu}_g^\top - N\hat{\mu}_g\hat{\mu}_g^\top = (N-1)\hat{S} + N\hat{\mu}_g\hat{\mu}_g^\top \tag{20}$$

It is a well known result that the global covariance can be expressed as:

$$\hat{S} = \frac{1}{N-1}\left(\sum_{c=1}^C (N_c - 1)\hat{\Sigma}_c + \sum_{c=1}^C N_c(\hat{\mu}_c - \hat{\mu}_g)(\hat{\mu}_c - \hat{\mu}_g)^T\right),$$

Replacing the global covariance $\hat{S}$ in Eq. (20), we obtain the final expression for $G$ as:

$$G = \sum_{c=1}^C (N_c - 1)\hat{S}_c + \sum_{c=1}^C N_c(\hat{\mu}_c - \hat{\mu}_g)(\hat{\mu}_c - \hat{\mu}_g)^\top + N\hat{\mu}_g\hat{\mu}_g^\top$$

$\square$

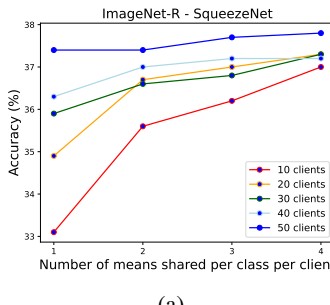 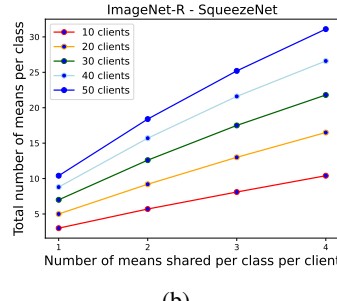

|(a)|(b)|

Figure 9: (a) Analysis of FedCOF performance with multiple class means per client on ImageNet-R. (b) Total number of means per class on average that are used to estimate the covariance for FedCOF in Fig. 9a.

## APPENDIX E    COMMUNICATION COSTS

When computing communication costs we consider that the pre-trained models are on the clients and do not need to be communicated. We do not include cost of backward communication of classifier parameters from server to clients, since it is the same for all methods but is necessary if the models are fine-tuned after classifier initialization. All parameters are considered to be 32-bit floating point numbers (i.e. 4 bytes).

## APPENDIX F    IMPLEMENTATION DETAILS

Here, we provide details on learning rate (lr) used for all fine-tuning experiments with FedAdam. For ImageNet-R, we use a lr of 0.0001 for both server and clients for FedNCM, Fed3R and FedCOF initializations. For CUB200 and Cars, we use a server lr of 0.00001 and client lr of 0.00005 for Fed3R and FedCOF, while for FedNCM, we use a higher lr of 0.0001 for clients.

For the linear probing experiments with FedAvg, we use a client lr of 0.01 and server lr of 1.0 for FedNCM across all datasets. For Fed3R and FedCOF initializations, we use a client lr of 0.001 and a server lr of 1.0 for all datasets. We use an Nvidia RTX 6000 GPU for our experiments.

## APPENDIX G    THE FEDCOF ORACLE (SHARING FULL COVARIANCES)

Similar to (Luo et al., 2021), we aggregate the class covariances from clients as follows:

$$\hat{\Sigma}_c = \sum_{k=1}^{K} \frac{n_{k,c} - 1}{N_c - 1} \hat{\Sigma}_{k,c} + \sum_{k=1}^{K} \frac{n_{k,c}}{N_c - 1} \hat{\mu}_{k,c} \hat{\mu}_{k,c}^T - \frac{N_c}{N_c - 1} \hat{\mu}_c \hat{\mu}_c^T. \tag{21}$$

We use the aggregated class covariance from Eq. (21) and apply shrinkage to obtain $\hat{\Sigma}_c + \gamma I_d$ and use it in Eq. (11) for the oracle setting of FedCOF.

## APPENDIX H    ADDITIONAL EXPERIMENTS

**Sampling multiple class means.**  We perform multiple class means sampling per client using ImageNet-R and show in Fig. 9a that using FedCOF with more class means shared from each client improves the performance. We also show in Fig. 9b the total number of means used per class on an average in Fig. 9a to perform the covariance estimation. The number of means used is less than the total number of clients due to the class-imbalanced or dirichlet distribution used to sample data for clients.

**Linear probing after initialization experiments.**  We show in Fig. 10 that linear probing after FedCOF classifier initialization improves the accuracy significantly compared to FedNCM and is marginally better than Fed3R initialization across three datasets using SqueezeNet.

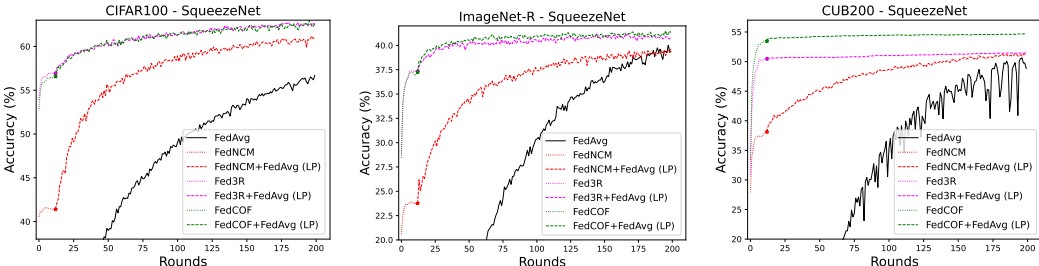

Figure 10: Analysis of the performance with linear probing using FedAvg (McMahan et al., 2017).

Table 3: Comparison of different training-free methods using SqueezeNet with training-based Fed-LP (federated linear probing with FedAvg (McMahan et al., 2017) starting with pre-trained model and random classifier initialization) across 5 random seeds. FedNCM, Fed3R and the proposed FedCOF does not involve any training. We show the total communication cost (in MB) from all clients to server. The best results from each section are highlighted in **bold**.

| Method | CIFAR100 | | ImageNet-R | | CUB200 | | CARS | | iNat-120K | |
|---|---|---|---|---|---|---|---|---|---|---|
| | Acc (↑) | Comm. (↓) | Acc (↑) | Comm. (↓) | Acc (↑) | Comm. (↓) | Acc (↑) | Comm. (↓) | Acc (↑) | Comm. (↓) |
| Fed-LP | **59.9**±0.2 | 2458 | **37.8**±0.3 | 4916 | 46.8±0.8 | 4916 | 33.1±0.1 | 4817 | 28.0±0.6 | $1.6 \times 10^6$ |
| FedNCM | 41.5±0.1 | **5.9** | 23.8±0.1 | **7.1** | 37.8±0.3 | **4.8** | 19.8±0.2 | **5.4** | 21.2±0.1 | **111.8** |
| Fed3R | 56.9±0.1 | 110.2 | 37.6±0.2 | 111.9 | 50.4±0.3 | 109.6 | 39.9±0.2 | 110.2 | 32.1±0.1 | 9837.3 |
| FedCOF (Ours) | 56.1±0.2 | **5.9** | **37.8**±0.4 | **7.1** | **53.7**±0.3 | **4.8** | **44.0**±0.3 | **5.4** | **32.5**±0.1 | **111.8** |

**Comparison of training-free methods with linear probing.** We also compare with our approach with the training-based federated linear probing without any initialization (where we perform FedAvg and learn only the classifier weights of models) and show in table Table 3 that FedCOF is more robust and communication-efficient compared to federated linear probing across several datasets. We follow the same settings as in Table 2. For first 4 datasets, we perform federated linear probing for 200 rounds with 30 clients per round using FedAvg with a client learning rate of 0.01. For iNat-120k, we train more for 5000 rounds.

**Experiments with ResNet18.** We perform experiments with pre-trained ResNet18 in table Table 4. For FedAvg and FedAdam, we train for 200 rounds with 30 clients per round. For FedAvg, we train with a client learning rate of 0.001 and server learning rate of 1.0. For FedAdam, we train with a client learning rate of 0.001 and a server learning rate of 0.0001.

**Impact of using pre-trained models.** To quantify impact of using pre-trained models we performed experiments using a randomly initialized model and show in Table 5 that federated training using a pre-trained model significantly outperforms a randomly initialized model using standard methods like FedAvg and FedAdam on CIFAR-10 and CIFAR-100.

## APPENDIX I CONVERGENCE ANALYSIS

In our work, we claim that FedCOF initialization achieves faster and better convergence based on our empirical results (Fig. 6) using multiple datasets with 100 classes (CIFAR100) and 200 classes (ImageNet-R, CUB200). We propose how to initialize the classifiers before performing federated optimization methods like FedAvg (McMahan et al., 2017) and FedAdam (Reddi et al., 2020) which have already established the theoretical guarantes of convergence in their respective works. Unlike gradient-based FL methods, our method is training-free. Similar to Fed3R (Fanì et al., 2024) and FedNCM (Legate et al., 2023a), the proposed FedCOF does not depend on assumptions like bounded variance of stochastic gradients or smoothness of clients objectives. While we do not propose any federated optimization step, we propose a training-free method that can be also used for initializing federated fine-tuning. We would also like to highlight that all existing works in Federated learning with pre-trained models (Tan et al., 2022b; Nguyen et al., 2020; Fanì et al., 2024; Legate et al., 2023a; Chen et al., 2022; Qu et al., 2022; Shysheya et al., 2022) focus only on empirical observations assuming that the theoretical guarantes of existing federated optimization methods holds true when using pre-trained models. A more exhaustive study on convergence analysis for FL with pre-trained models would be an interesting direction to explore in future works.

Table 4: Comparison of different training-free methods using pre-trained ResNet18 for 100 clients with training-based federated learning baselines FedAvg (McMahan et al., 2017) and FedAdam (Reddi et al., 2020) starting from a pre-trained model. We train for 200 rounds for FedAvg and FedAdam. FedNCM, Fed3R and the proposed FedCOF do not involve any training. We also show the performance of fine-tuning with FedAdam after classifier initialization. For fine-tuning experiments we only train for 100 rounds after initialization. We show the total communication cost (in MB) from all clients to server. The best results from each section are highlighted in **bold**.

| | CIFAR100 | | ImageNet-R | |
|---|---|---|---|---|
| Method | Acc ($\uparrow$) | Comm. ($\downarrow$) | Acc ($\uparrow$) | Comm. ($\downarrow$) |
| FedAvg | 67.7 | 538k | 56.0 | 541k |
| FedAdam | 74.4 | 538k | 57.1 | 541k |
| FedNCM | 53.8 | **5.9** | 37.2 | **7.1** |
| Fed3R | 63.5 | 110.2 | 45.9 | 111.9 |
| FedCOF (Ours) | 63.3 | **5.9** | 46.4 | **7.1** |
| FedNCM+FedAdam | 75.7 | 269k | 60.3 | 271k |
| Fed3R+FedAdam | 76.8 | 269k | 60.6 | 271k |
| FedCOF+FedAdam | **76.9** | 269k | **62.2** | 271k |

Table 5: Impact of using pre-trained SqueezeNet network with different federated learning methods on CIFAR100. We show the total communication cost (in MB) from all clients to server. We train 100 clients with 30 clients per round for 200 rounds in non-iid settings with dirichlet distribution of 0.1. When starting from random initialization (no pre-training), we train for 400 rounds.

| | | CIFAR10 | | CIFAR100 | |
|---|---|---|---|---|---|
| Method | Pre-trained | Acc ($\uparrow$) | Comm. ($\downarrow$) | Acc ($\uparrow$) | Comm. ($\downarrow$) |
| FedAvg | $\times$ | 37.3 | 74840 | 23.9 | 79248 |
| FedAdam | $\times$ | 60.5 | 74840 | 44.3 | 79248 |
| FedAvg | $\checkmark$ | 84.7 | 37420 | 56.7 | 39624 |
| FedAdam | $\checkmark$ | 85.5 | 37420 | 62.5 | 39624 |

## APPENDIX J  BIAS OF THE ESTIMATOR WITH NON-IID CLIENT FEATURES

In Appendix C we showed that, under the assumption that the per-class features are iid across clients, the proposed estimator is an *unbiased estimator*. In this section, we theoretically quantify the bias when the i.i.d assumption is violated.

Under the i.i.d. assumption, the single class features assigned to clients can be treated as random samples from the *same* population distribution with mean $\mu_c$ and covariance $\Sigma_c$. For simplicity, focusing on a single class and dropping the class subscript $c$, the population distribution has mean $\mu$ and covariance $\Sigma$. As a result, recalling equation 16, we can write:

$$\mathbb{E}[\overline{F}_k \overline{F}_k^\top] = \mathrm{Var}[\overline{F}_k] + \mathbb{E}[\overline{F}_k]\mathbb{E}[\overline{F}_k]^\top = \frac{\Sigma}{n_k} + \mu\mu^\top,$$

where $n_k$ is the number of samples assigned to client $k$, and $\overline{F}_k$ is the sample mean for client $k$

Now, if the *i.i.d assumption is violated* the local features assigned to each client can be viewed as random samples drawn from different client population distributions, each characterized by a mean $\mu_k$ and covariance $\Sigma_k$, with $\mu_i \neq \mu_j$ and $\Sigma_i \neq \Sigma_j$ for $i \neq j$, and $i, j = 1, \ldots, K$. In this case:

$$\mathbb{E}[\overline{F}_k \overline{F}_k^\top] = \mathrm{Var}[\overline{F}_k] + \mathbb{E}[\overline{F}_k]\mathbb{E}[\overline{F}_k]^\top = \frac{\Sigma_k}{n_k} + \mu_k\mu_k^\top. \tag{22}$$

To compute the expectation of the estimator $\mathbb{E}[\hat{\Sigma}]$, we follow the same procedure used to prove proposition in Appendix C up to Eq. 15:

$$\mathbb{E}[\hat{\Sigma}] = \frac{1}{K-1}\left(\sum_{k=1}^{K} n_k \mathbb{E}[\overline{F}_k \overline{F}_k^\top] - 2N\mathbb{E}[\hat{\mu}\hat{\mu}^\top] + \sum_{k=1}^{K} n_k \mathbb{E}[\hat{\mu}\hat{\mu}^\top]\right). \tag{23}$$

Assuming the global feature dataset, regardless of client assignment, is a random sample from the population with mean $\mu$ and covariance $\Sigma$, we can write:

$$\mathbb{E}[\hat{\mu}\hat{\mu}^\top] = \mathrm{Var}[\hat{\mu}] + \mathbb{E}[\hat{\mu}]\mathbb{E}[\hat{\mu}]^\top = \frac{\Sigma}{N} + \mu\mu^\top. \tag{24}$$

Table 6: Comparison of different training-free methods using MobileNetV2 on the feature shift setting on DomainNet. We show the total communication cost (in MB) from all clients to server.

| Method | Acc (↑) | Comm. (↓) |
|---|---|---|
| FedNCM | 65.8 | 0.3 |
| Fed3R | 81.9 | 39.6 |
| FedCOF | 74.1 | 0.3 |
| FedCOF (2 class means per client) | 76.5 | 0.6 |
| FedCOF (10 class means per client) | 78.8 | 3.1 |

Substituting Equations 24 and 22 into Equation 23, and recalling that $N = \sum_{k=1}^{K} n_k$, we obtain:

$$\mathbb{E}[\hat{\Sigma}] = \frac{1}{K-1} \left( \sum_{k=1}^{K} n_k(\frac{\Sigma_k}{n_k} + \mu_k\mu_k^\top) - 2N(\frac{\Sigma}{N} + \mu\mu^\top) + \sum_{k=1}^{K} n_k(\frac{\Sigma}{N} + \mu\mu^\top) \right)$$

$$= \frac{1}{K-1} \left( \sum_{k=1}^{K} n_k(\frac{\Sigma_k}{n_k} + \mu_k\mu_k^\top) - \Sigma - N\mu\mu^\top \right)$$

$$= \frac{1}{K-1} \sum_{k=1}^{K} (\Sigma_k - \frac{\Sigma}{K}) + \frac{1}{K-1} \left( \sum_{k=1}^{K} n_k\mu_k\mu_k^\top - \sum_{k=1}^{K} n_k\mu\mu^\top \right)$$

$$= \frac{1}{K-1} \sum_{k=1}^{K} (\Sigma_k - \frac{\Sigma}{K}) + \frac{1}{K-1} \sum_{k=1}^{K} n_k(\mu_k\mu_k^\top - \mu\mu^\top)$$

$$= \frac{1}{K-1} \sum_{k=1}^{K} (\Sigma_k - \frac{\Sigma}{K}) + \frac{1}{K-1} \sum_{k=1}^{K} n_k(\mu_k - \mu)(\mu_k - \mu)^\top,$$

where in the last step we used that $\sum_{k=1}^{K} n_k\mu_k = N\mu$.

The bias of the estimator is thus given by:

$$\text{Bias}(\hat{\Sigma}) = \mathbb{E}[\hat{\Sigma}] - \Sigma = \frac{1}{K-1} \sum_{k=1}^{K} (\Sigma_k - \Sigma) + \frac{1}{K-1} \left( \sum_{k=1}^{K} n_k(\mu_k - \mu)(\mu_k - \mu)^\top \right). \quad (25)$$

Note that if each client population covariance $\Sigma_k$ is equal to the global population covariance $\Sigma$, and the mean of each client $\mu_k$ is equal to the population mean, then the bias is zero (i.e., the estimator is unbiased). However, the bias formula reveals that when the distribution of a class within a client differs from the global distribution of the same class, our estimator introduces a systematic bias. This situation can arise in the *feature-shift* setting, in which each client is characterized by a different domain. In the next section, we evaluate FedCOF under the feature-shift setting to quantify how this bias affects performance in this specific scenario.

As a final note, we mention that we always assume the global distribution of a single class can be modeled with a distribution having a single mean and covariance (see Eq. 24). This is how our classifier operates. As future work, it could be beneficial to employ different types of classifiers that allow multiple class means and class covariances.

## APPENDIX K  EXPERIMENTS ON FEATURE SHIFT SETTINGS

Following (Li et al., 2021), we perform experiments with MobileNetv2 in a non-iid feature shift setting on the DomainNet (Peng et al., 2019) dataset. DomainNet contains data from six different domains: Clipart, Infograph, Painting, Quickdraw, Real, and Sketch. We use the top 10 most common classes of DomainNet for our experiments following the setting proposed by (Li et al., 2021). We consider six clients where each client has i.i.d. data from one of the six domains. As a result, different clients have data from different feature distributions. We show in Table 6 how training-free methods perform in feature shift settings and the accuracy to communication trade-offs.

Fed3R achieves better overall performance then FedCOF, likely due to its use of exact class covariance, avoiding the bias that FedCOF introduces. However, FedCOF achieves comparable results

while significantly reducing communication costs. FedNCM perform worse than FedCOF at the same communication budget. When we increase the number of means sampled from each client, the performance of our approach improves. This is due to the fact that our method suffers with low number of clients (only 6 in this experiments) and sampling multiple means helps.

## APPENDIX L   PRIVACY CONCERNS ON SHARING CLASS-WISE STATISTICS

Our method requires transmitting class-wise statistics to compute the unbiased estimator of the population covariance (Eq. 14) and classifier initialization, similar to other methods in federated learning (Legate et al., 2023a; Luo et al., 2021). In general, transmitting the class-wise statistics may raise privacy concerns, since each client could potentially expose its class distribution. Inspired by differential privacy (Dwork et al., 2006), we propose perturbing the class-wise statistics of each client with different types and intensities of noise, before transmission to the global server. This analysis allows us to evaluate how robust FedCOF is to variations in class-wise statistics and whether noise perturbation mechanisms can effectively hide the true client class statistics. Specifically, we propose perturbing the class-wise statistics as follows:

$$\widetilde{n}_{k,c} = \max(n_{k,c} + \sigma_\epsilon^{\text{noise}}, 0) \tag{26}$$

where $\sigma_\epsilon^{\text{noise}}$ is noise added to the statistics, and $\epsilon$ is a parameter representing the noise intensity. The $\max$ operator clips the class statistics to zero if the added noise results in negative values, which is expected to happen in federated learning with highly heterogeneous client distributions. When clipping is applied, the client does not send the affected class statistic and class mean, and the server excludes them from the computation of the unbiased estimator.

We consider three types of noise:

- *Uniform noise*: $\sigma_\epsilon^{\text{unif}} \sim \mathcal{U}(-(1 - \epsilon)n_{k,c}, +(1 - \epsilon)n_{k,c})$, proportional to the real class statistics.

- *Gaussian noise*: $\sigma_\epsilon^{\text{gauss}} \sim \mathcal{N}(0, \frac{1}{\epsilon})$, independent of the real class statistics.

- *Laplacian noise* $\sigma_\epsilon^{\text{laplace}} \sim \mathcal{L}(0, \frac{1}{\epsilon})$, which is also independent of the real class statistics.

Lower $\epsilon$ values correspond to higher levels of noise in the statistics.

In Figure 11, we show that the performance of FedCOF is robust with respect to the considered noise perturbation, varying the intensity of $\epsilon \in \{0.1, 0.3, 0.5, 0.7, 0.9\}$. These results suggest that a differential privacy mechanism can be implemented to mitigate privacy concerns arising from the exposure of client class-wise frequencies. In Figure 12, we provide a qualitative overview of how the proposed Laplacian and uniform noise perturbation affect class-wise distributions.

## APPENDIX M   DATASET DETAILS

Apart from CIFAR100, we perform experiments with ImageNet-R (Hendrycks et al., 2021) which is an out-of-distribution dataset and proposed to evaluate out-of-distribution generalization using ImageNet pre-trained weights. It contains data with multiple styles like cartoon, graffiti and origami which is not seen during pre-training. We also consider fine-grained datasets like CARS and CUB200 for our experiments. Finally, we also use iNaturalist-Users-120k (Hsu et al., 2019) dataset in our experiments, which is a real-world, large-scale dataset (Van Horn et al., 2018) proposed by (Hsu et al., 2019) for federated learning and contains 120k training images of natural species taken by citizen scientists around the world, belonging to 1203 classes spread across 9275 clients. In datasets like ImageNet-R and CARS, we also face class-imbalanced situations where there is a significant class-imbalance at the global level.

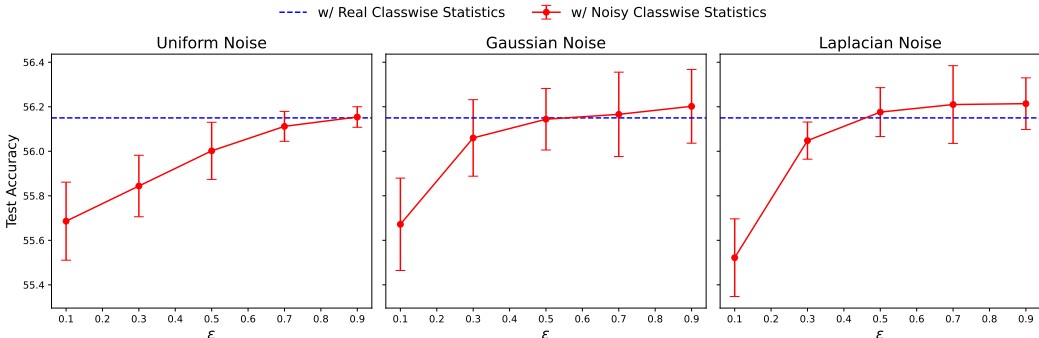

Figure 11: Performance of FedCOF with noisy class statistics on CIFAR-100 using SqueezeNet. The number of clients is fixed at 100 and classes are distributed using a Dirichlet distribution with $\alpha = 0.1$. Results are averaged over five random seeds, each generating different noise in client statistics, and the standard deviation is reported. FedCOF demonstrates robustness to uniform, Gaussian, and Laplace perturbations in class statistics, with performance showing a slight drop as noise, parameterized by $\epsilon$, increases. Lower $\epsilon$ corresponds to higher noise levels in the class statistics.

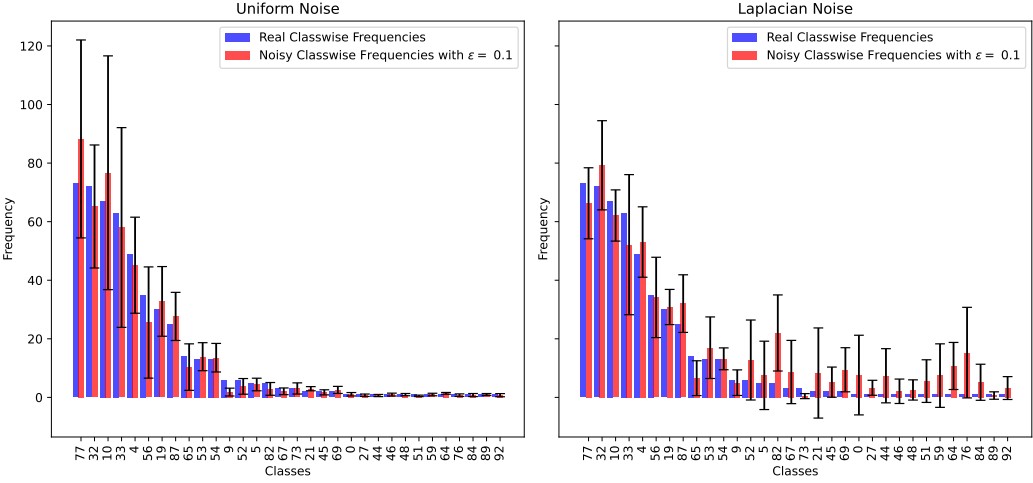

Figure 12: Class frequency distributions for a single client under different noise types: uniform noise (left) and Laplacian noise (right) on CIFAR-100. Both noise types are applied to the real class statistics with the highest noise intensity ($\epsilon = 0.1$). The bar heights represent the average class frequencies, and the error bars indicate the standard deviation across 5 seeds. Real class-wise frequencies and their noisy counterparts are shown for comparison.

