# OpenReview forum: "Covariances for Free: Exploiting Mean Distributions for Federated Learning with Pre-trained Models"
_ICLR.cc/2025/Conference — Submitted to ICLR 2025_

### Official Review · Reviewer_6242 · 2024-10-30

**Soundness:** 3
**Presentation:** 3
**Contribution:** 2
**Rating:** 6
**Confidence:** 3

**Summary:**

This paper proposes a novel training-free FL method called FedCOF, which approximates the covariance on the server side to eliminate the enormous communication overhead. Numerical results demonstrate that FedCOF achieves comparable performance to Fed3R by merely transmitting class means to the server.

**Strengths:**

1. Building on theoretical guarantees, this paper introduces a novel algorithm that eliminates the need for transmitting covariances between the server and clients while maintaining performance levels. This represents a valuable step for training-free federated learning.

2. The authors have carried out extensive experiments to validate the effectiveness of the proposed method, demonstrating considerable effort in their research.

**Weaknesses:**

While the motivation of this paper is clear, I have the following questions/discussions.

1. The algorithm necessitates the transmission of $n_{k,c}$ to the server, which introduces certain privacy concerns. Although other methods also require this information, it would be beneficial if the authors could discuss potential techniques to address or mitigate this issue.

2. As modern pre-trained models tend to be generative models (e.g., GPT), it would be interesting to explore the possibility of extending the proposed methods to handle generative models by initializing the decoding heads accordingly.

**Questions:**

Please see weakness.

---

> ### Author Response · Authors · 2024-11-21
>
> We thank the reviewer for their kind works about our work. We greatly appreciate your recognition that our contribution  represents a valuable step for training-free federated learning.  We are also grateful for your acknowledgment of the considerable effort we put into our research, including the extensive experiments validating the effectiveness of our proposed method. Below, we address the questions and comments you raised:
>
> >The algorithm necessitates the transmission of n_{k,c} to the server, which introduces certain privacy concerns. Although other methods also require this information, it would be beneficial if the authors could discuss potential techniques to address or mitigate this issue.
>
> We agree with the reviewer that sharing the class statistics introduces certain privacy concerns. Following this suggestion (and the concern raised by the  Reviewer [jBiQ](https://openreview.net/forum?id=7NtAIghBsE&noteId=5ZSh899J9u)) we decided to investigate methodologies to mitigate these privacy concerns. We propose perturbing class-wise statistics with different types and intensities of noise before transmitting them to the global server and evaluate the performance of FedCOF. Specifically, we perturb the class-wise statistics as follows:
> $$ñ_{k,c}= \max(n_{k,c} + \sigma^{\text{noise}}_{\epsilon},0).$$
>
> Here $\sigma^{\text{noise}}_{\epsilon}$ represents noise with intensity parametrized by $\epsilon$. The $\max$ operator ensure non-negative values in client statistics.
>
> We vary the intensity of $\epsilon$ and the type of noise applied, including Uniform, Gaussian and Laplacian noise. Our findings demonstrate that the performance of FedCOF is robust to the noise types with varying intensities. These results suggest that perturbing statistics can mitigate privacy concerns stemming from the exposure of client class-wise frequencies. We added this empirical analysis in the Supplementary Material (Appendix L).
>
> >As modern pre-trained models tend to be generative models (e.g., GPT), it would be interesting to explore the possibility of extending the proposed methods to handle generative models by initializing the decoding heads accordingly.
>
> We thank the reviewer for raising this interesting question. In principle, we think that initializing the linear layer that maps tokens to logits in autoregressive generative models would be possible. However, it is not clear whether the evaluated training-free appraoches proposed for federated learning including FedCOF would generalize to generative, autogressive  settings. We think this could be an interesting direction for a future work.

---

> > ### Comment · Reviewer_6242 · 2024-11-25
> > **Reply to authors**
> >
> > I thank the authors for their detailed rebuttal and the additional experiments. My primary concerns have been addressed, and I am now inclined to maintain my positive recommendation.

---

### Official Review · Reviewer_z1Dd · 2024-10-31

**Soundness:** 4
**Presentation:** 3
**Contribution:** 3
**Rating:** 8
**Confidence:** 4

**Summary:**

The paper tackles training-free federated learning. It proposes to estimate the per-class covariances in a federated learning setting using the variance of local per-class means. This covariance can be used to obtain a ridge regression classifier, which outperforms a nearest-neighbor classifier based on class means. The proposed approach thereby avoids sending local covariance matrices which reduces communication and potential privacy risks.

The derivation of the estimator for the per-class covariance is sound. The empirical evaluation is comprehensive.

**Strengths:**

- Training-free federated learning using feature extractors is a relevant and interesting problem.
- The proposed estimator of covariances is sound and novel.
- Experiments show substantial improvement over existing training-free methods and potential for its combination with federated fine-tuning or linear probing.

**Weaknesses:**

- The impact of the iid assumption on realistic scenarios is evaluated empirically. It would be great to quantify how heterogeneous distributions impact the estimator theoretically, e.g., under the assumptions that local distributions are Gaussians with different mean or covariance.
- The dimension of the feature space could impact the accuracy of the estimator. This impact should be evaluated, e.g., using a synthetic dataset with varying number of feature dimensions.
- The paper focusses on label shifts, i.e., a heterogeneous distribution of classes. It is unclear how the method performs in case of feature shift, i.e., a heterogeneous distribution of features, e.g., via locally different covariance structures [1].

**Questions:**

- Why is this approach better in terms of accuracy that Fed3R? Shouldn't it perform slightly worse or en-par, since it only approximates the covariance matrix? Here it would be good to investigate the approximation of the covariance matrix and compare it to the one produced by Fed3R.
- It would be great to compare the results using a strong pre-trained feature extractor with a classical end-to-end federated learning baseline, e.g., training a ResNet-50 on CIFAR100.
- For consistency I suggest to use $\widehat{\Sigma}$ instead of $\widehat{S}$ in Eq. 10.
- Please compare your approach also to distributed training of linear models (using standard FedAvg), since ideally the training-free approach should perform at least en-par in terms of model performance and should outperform them in terms of communication. Here, it would be particularly interesting to compare to communication-efficient approaches [2].
- Since ridge regression might not always be ideal approach given a fixed feature extractor, I wonder whether a kernel ridge regression could be applicable. This would require sending the kernel matrix, but also has a closed form solution. The communication cost in that case would be quadratic in the number of data points, rather then linear in the features, so for many scenarios communication might be higher. Once could employ compression techniques here, though, like the Nyström method.


[1] Li, Xiaoxiao, et al. "FedBN: Federated Learning on Non-IID Features via Local Batch Normalization." International Conference on Learning Representations, 2021.
[2] Kamp, Michael, et al. "Communication-efficient distributed online prediction by dynamic model synchronization." Machine Learning and Knowledge Discovery in Databases, 2014.

---

> ### Author Response · Authors · 2024-11-21
> **Response 1/2**
>
> We greatly appreciate the acknowledgment that we address a relevant and interesting problem, and we are pleased by the recognition that our proposed unbiased estimator for estimating class covariances in federated learning is sound, novel, and capable of achieving substantial improvement over existing training-free methods. Below, we address specific concerns raised by the reviewer.
>
> > The impact of the iid assumption on realistic scenarios is evaluated empirically. It would be great to quantify how heterogeneous distributions impact the estimator theoretically, e.g., under the assumptions that local distributions are Gaussians with different mean or covariance.
>
> Many thanks to the reviewer for this insightful question. To theoretically quantify how heterogenous distributions impact our estimator $\hat{\Sigma}$, we derive a general bias formula independent of the i.i.d assumption. By treating each client as a random sample from distinct population distributions with mean $\mu_k$ and covariance $\Sigma_k$, we repeat the calculus for proving Proposition 2 in Appendix C. After some numerical steps, the general bias formula we derive is:
> $$\text{Bias}(\hat{\Sigma}) = \text{E}[\hat{\Sigma}] -  \Sigma = \frac{1}{K-1} \sum_{k=1}^K (\Sigma_k - \Sigma) + \frac{1}{K-1}\left(\sum_{k=1}^K n_k (\mu_k -\mu )(\mu_k -\mu)^\top \right),$$ where $n_k$ is the number of samples assigned to a client , $\mu$ and $\Sigma$ are the global mean and covariance of all the features independently by the client assignement. Here as previously done, we are focusing on a single class.
>
> If each client population covariance $\Sigma_k$ equals to the global covariance $\Sigma$, and the client mean $\mu_k$ matches the global mean $\mu$, the bias is zero, making the estimator unbiased. However, the bias formula shows that if a class distribution within a client differs from the global distribution of the same class, the estimator introduces a systematic bias. This situation can arise in the *feature-shift* setting, in which each client is characterized by a different domain. Quantifying this bias can open future directions for designing estimators that account for highly heterogenous distribution.
> In the Supplementary Material (Appendix J), we provide the mathematical derivation to arrive at this general formula for the bias of our estimator and this discussion.
>
> >The paper focuses on label shifts, i.e., a heterogeneous distribution of classes. It is unclear how the method performs in case of feature shift, i.e., a heterogeneous distribution of features, e.g., via locally different covariance structures [1].
>
> Now, we empirically evaluate the performance of FedCOF in the feature-shift setting to understand how much the bias affect the performance in this setting. For this setting we use the DomainNet dataset, with six different domains. Following the work [1] suggested by the reviewer, we consider six clients where each client has i.i.d. data from one of the six domains.
>
> | Method| Acc (↑) | Comm. (↓) |
> |:--:|:--:|:--:|
> | FedNCM |65.8|0.3|
> |Fed3R | 81.9|39.6 |
> | FedCOF| 74.1| 0.3 |
> | FedCOF (2 class means per client)  | 76.5 | 0.6 |
> | FedCOF (10 class means per client) | 78.8 |3.1 |
>
> Fed3R achieves better overall performance then FedCOF, likely due to its use of real class covariance, avoiding the bias addition that FedCOF introduces. However, FedCOF achieves comparable results while significantly reducing communication costs. FedNCM perform worse than FedCOF, at the same communication budget. When we increase the number of means sampled from each client, the performance of our approach improves. This is due to the fact that our method suffer with low number of clients (only 6 in this experiments) and sampling multiple means helps, as mentioned in the main paper. In Supplementary Material (Appendix K), we add these experiments and discussion.
>
> > The dimension of the feature space could impact the accuracy of the estimator.
>
> We already analyzed how different feature space dimensionalities can affect performance (512 for SqueezeNet and ResNet18, 1280 for MobileNetV2 and 768 for ViT-B/16) but these are also dependent on the quality of the features and thus depends on the network architecture. We agree that very high-dimensional features can affect estimator accuracy due to low-rank, high-dimensional covariance estimates from limited samples (L273-284). We mitigate this issue by using covariance shrinkage regularization which adds a multiple of an identity matrix to the covariance estimate. Hoewever, we have not yet found a way to construct a representative example with gaussians to illustrate the impact of the dimensionality. We will expand the discussion on high dimensional synthetic features and plan to provide an example with synthetic dataset in the final version.
>
> >For consistency I suggest to use $\widehat{\Sigma}$ instead of $\widehat{S}$ in Eq. 10.
>
> In the final version of the paper, we will fix this.

---

> ### Author Response · Authors · 2024-11-21
> **Response 2/2**
>
> >Why is this approach better in terms of accuracy that Fed3R?
>
> We would like to highlight that the proposed method FedCOF has a different classifier compared to Fed3R which uses a ridge regression classifier. We discussed this in section 4.3. The classifier initialization of Fed3R uses $G$ obtained from Equation 10, which considers both the within- and between- class scatter matrices. We propose a different classifier initialization using Equation 11, which uses **only within-class scatter matrices**.
>
> We empirically motivate this by analyzing the impact of within class scatter matrices in Figure 4. Using a centralized setting, we showed that classification performance can be improved by removing the between-class covariances from Equation 10. So, in FedCOF we initialize the classifier using $G$ from Equation 11 which uses *only* the estimated within-class covariances and thus performs better than Fed3R in most settings. Thus, the improvement in performance of FedCOF compared to Fed3R is due to the different classifier initialization.
>
> Regarding the covariance approximation, FedCOF-Oracle uses true class covariances while FedCOF uses the proposed estimator using client means. While both employ the same formula for classifier initialization (equation 11), FedCOF-Oracle generally outperforms FedCOF because the rank of our estimated covariance is limited by the number of clients per class (see Section 4.2 "Impact of the Number of Clients"), resulting in a lower-rank approximation compared to the true class covariance.
>
> > Strong pre-trained feature extractor with a classical end-to-end federated learning baseline.
>
> We perform experiments with ResNet-18 instead of ResNet50 due to limited time and compute resources and show how the classical federated learning approaches perform. We observe significant improvement on using FedCOF and finetuning with FedAdam compared to simply using FedAdam. We discuss this in details in Appendix H (Experiments with ResNe18, see Table 4). We also had the comparison with FedAdam in Figure 6 of our paper on three datasets using SqueezeNet architecture where we show that FedCOF+FedAdam outperforms FedAdam.
>
> |||CIFAR100||IN-R|
> |:-:|:-:|:-:|:-:|:--:|
> |Method|Acc. (↑)|Comm. (in MBs) (↓)|Acc. (↑)|Comm. (in MBs) (↓)|
> |FedAvg|67.7|538k|56.0|541k|
> |FedAdam|74.4|538k|57.1|541k|
> |FedNCM|53.8|5.9|37.2|7.1|
> |Fed3R|63.5|110.2|45.9|11.9|
> |FedCOF| 63.3 | 5.9 | 46.4 | 7.1 |
> |FedNCM+FedAdam| 75.7 | 269k | 60.3 | 271k |
> |Fed3R+FedAdam| 76.8 | 269k | 60.6 | 271k |
> |FedCOF+FedAdam| 76.9 | 269k  | 62.2 | 271k |
>
> >Please compare your approach also to distributed training of linear models (using standard FedAvg).
>
> We now compare with our approach with the training-based federated linear probing (where we perform FedAvg and learn only the classifier weights of models) and show in table below that FedCOF is more robust and communication-efficient compared to federated linear probing across several datasets. We discuss this in details in Appendix H (Comparison of training-free methods with linear probing, Table 3).
>
> |||CIFAR100||IN-R||CUB200||CARS||iNat-120k|
> |:--:|:--:|:--:|:--:|:--:|:--:|:--:|:--:|:--:|:--:|:--:|
> | Method | Acc.  (↑)| Comm. (↓) | Acc. (↑) | Comm. (↓) | Acc. (↑) | Comm. (↓) | Acc. (↑) | Comm. (↓) |  Acc. (↑) | Comm. (↓) |
> | Fed-LP | 59.9 $\pm$ 0.2 | 2458  | 37.8 $\pm$ 0.3 | 4916 | 46.8 $\pm$ 0.8 | 4916 | 33.1 $\pm$ 0.1 | 4817 | 28.0 $\pm$ 0.6 | 1.6 $\times$ 10^6 |
> | FedCOF (ours) | 56.1 $\pm$ 0.2 | 5.9 | 37.8 $\pm$ 0.4 | 7.1| 53.7 $\pm$ 0.3 | 4.8 | 44.0 $\pm$ 0.3 | 5.4 | 32.5 $\pm$ 0.1 | 111.8 |
>
> >Since ridge regression might not always be ideal approach given a fixed feature extractor, I wonder whether a kernel ridge regression could be applicable.
>
> We agree that using a fixed feature extractor and employing ridge regression may not be optimal since the features might not be well separated. Kernel ridge regression could address this by implcitly mapping features to a higher dimensional space and thus improving separability. However, as noted this would increase communication costs quadratically with the number of data points and employing the Nystrom method, could mitigate these additional communication costs and we will certainly consider this possibility for future work.
>
> Currently we see the following challenges:
> 1. Selecting kernel hyperparameters  (e.g $\sigma$ for the kernel RBF), is dataset dependent, determining a fixed $\sigma$ across diverse dataset is very challenging.
> 2. High-dimensional feature spaces in Kernel Ridge Regression amplify the need for careful shrinkage tuning to stabilize smaller eigenvalues, as mentioned in our paper. An automatic shrinkage estimation technique may help in this scenario.
> 3. The Nystrom method requires selecting $m$ samples from each client, where $m<n$, with $n$ being the total number of data points. How to choose an appropriate $m$ and determining which $m$ samples to select is not obvious in a highly heterogeneous federated setting.

---

> > ### Comment · Reviewer_z1Dd · 2024-11-22
> > **Response to authors**
> >
> > I want to thank the authors for the detailed reply. The derivation of the bias for heterogeneous local distributions is a great addition to the paper. The additional experiments show that the proposed method is sound and competitive. I maintain my positive rating.

---

### Official Review · Reviewer_Wumb · 2024-11-01

**Soundness:** 3
**Presentation:** 3
**Contribution:** 3
**Rating:** 6
**Confidence:** 2

**Summary:**

The paper proposes to use pre-trained models to perform federated classification. More specifically, Fed-COF uses pre-trained models to extract features of each example on the client, then averages the features within each class. The class-averaged features from clients are aggregated in the server to estimate the first-order and second-order statistics, which are then used in ridge regression to fit a classifier. The communication cost of Fed-COF scales only linearly with the size of the embedding.

**Strengths:**

The paper is well written. The derivations are clear and easy to understand. The proposed Fed-COF achieves decent empirical performance. Fed-COF also seems to work well with fine-tuning.

**Weaknesses:**

- The steps of Fed-COF + Fine-tuning can be made clearer. The description around line 357 is difficult to follow.

- The choice of ridge regularization parameter $\lambda$ seems important for the classification performance. Can authors give more empirical suggestions on how $\lambda$ should change with different numbers of clients/means per client?

**Questions:**

- What is the difference between Fed-COF oracle and Fed3R? In Table 2, it is surprising to see Fed3R sometimes achieves lower accuracy with more communication. The authors should provide more explanations for the phenomenon.

- In line 472, atleast should be at least.

---

> ### Author Response · Authors · 2024-11-21
>
> We thank the reviewer for appreciating the writing quality of our manuscript, the clarity on our theoretical derivation, and for recognizing that our approach demonstrates decent empirical performance for both classifier initialization and federated fine-tuning. Below we reply to the specific comments made.
>
> >The steps of Fed-COF + Fine-tuning can be made clearer. The description around line 357 is difficult to follow.
>
> Regarding FedCOF+Fine-tuning, we have more discussions in section 5.2 in the paper. In L357, we discuss how FedCOF classifier initialization can be used in multiple rounds before any finetuning starts. We consider the realistic setting when all the clients are not available at the same time and only a fraction of clients comes in each round. While FedNCM assumes the availability of all clients in one round for classifier initialization, we follow the multi-step approach proposed by Fed3R. We now clarify and update the discussion on FedCOF multiple rounds (L357-367).
>
> >The choice of ridge regularization parameter seems important for the classification performance. Can authors give more empirical suggestions on how lambda should change with different numbers of clients/means per client?
>
> Following Fed3R, we use the ridge regularization parameter to ensure that the matrix $G$ is invertible. This is only for numerical stability purposes. We performed an ablation to measure the impact of $\lambda$ and observe that the performance does not vary much (we notice a deviation of 0.1 on CIFAR100 and 0.25 on CUB200 by varying $\lambda$ from 0.001 to 1). In cases when the matrix $G$ is low-rank, such as in Fed-3R, the $\lambda$ parameter is helpful since it makes $G$ invertible. However, in our method we are not so dependent on $\lambda$ since we already use covariance shrinkage to obtain full-rank estimates of the covariance matrix (see equation 8) and as a consequence, we obtain an invertible matrix $G$ (see Equation 11).
>
> >What is the difference between Fed-COF oracle and Fed3R? In Table 2, it is surprising to see Fed3R sometimes achieves lower accuracy with more communication. The authors should provide more explanations for the phenomenon
>
> We would like to highlight that the proposed method FedCOF has a different classifier compared to Fed3R which uses a ridge regression classifier. We discussed this in section 4.3. The classifier initialization of Fed3R uses $G$ obtained from Equation 10, which considers both the within- and between- class scatter matrices. We propose a different classifier initialization using Equation 11, which uses **only within-class scatter matrices**.
>
> We empirically motivate this by analzing the impact of within class scatter matrices in Figure 4. Using a centralized setting, we showed that classification performance can be improved by removing the between-class covariances from Equation 10. So, in FedCOF we initialize the classifier using $G$ from Equation 11 which uses *only* the estimated within-class covariances and thus performs better than Fed3R in most settings. The FedCOF oracle uses the same classifier initialization as FedCOF but using the real covariances shared by clients. Thus, the difference between FedCOF-oracle and Fed3R is due to the different classifier initialization.

---

### Official Review · Reviewer_jBiQ · 2024-11-05

**Soundness:** 2
**Presentation:** 2
**Contribution:** 1
**Rating:** 3
**Confidence:** 4

**Summary:**

The authors introduce FedCOF, a training-free federated learning approach that utilizes a pretrained model's feature extractor while updating only the classifier. Unlike previous methods that only aggregate global means, FedCOF improves performance by also deriving and leveraging unbiased global covariances from these means. Local clients send first-order statistics (class-wise feature means) to the server, which then uses these to estimate global covariances. This innovation allows for efficient communication while significantly enhancing the effectiveness of the global classifier updates.

**Strengths:**

**Strengths of  FedCOF:**

FedCOF presents a timely contribution that leverages pretrained models in federated learning (FL), addressing the deep learning (DL) field's growing emphasis on foundation models. FedNCM is communication-efficient but suffers from limited performance, while Fed3R improves performance using both first- and second-order statistics but at a high communication cost. In contrast, FedCOF achieves strong performance with minimal communication overhead by deriving unbiased global covariances using only first-order statistics.

**Weaknesses:**

***Concerns on Presentation***

**Related Works**
I recommend enhancing the Related Works section to include a broader range of studies that cover both the use of fixed classifiers and the potential limitations of pretrained models in federated learning settings.

In **Line L101**, the section discussing the application of **fixed classifiers** in federated learning could be expanded by incorporating recent and relevant studies. Specifically, it would be valuable to reference works such as FedBABU [1], SphereFed [2], and Neural Collapse-inspired approaches [3,4], which explore the impact of classifier freezing in federated scenarios.

Furthermore, the discussion in **Line L102** and beyond about **federated learning with pretrained models** should present a more balanced view. The current description highlights only the positive outcomes of using pretrained models. However, it is important to acknowledge that pretrained models are not always advantageous in federated settings. For example, findings from FedFN [5], particularly in Section 5.2, demonstrate situations where pretrained models can adversely affect the performance of the global model, especially under heterogeneous data conditions. Including this perspective would provide a more comprehensive understanding of the complexities involved in using pretrained models within federated learning frameworks.

**Preliminaries**

L117: D_k seems to refer to the local dataset rather than local data.

L127: There is no clarification on the type of loss function or how the loss is calculated (whether as a batch mean or batch sum).

L129-130: "After initializing \theta with pretrained weights, the models can be optimized in a federated manner" — In this paper, local clients do not perform local updates based on the pretrained model, and this information seems to hinder the understanding of the paper.

**Concerns on Privacy Discussion in Section 4:**

The algorithm sends the class-wise frequency of the data held by clients to the central server. I believe this information could also raise privacy concerns, yet there is no mention of this issue. In fact, there are many previous FL papers that have communicated class frequency information and provided justifications. Citing these studies would strengthen the discussion, but this type of content is entirely missing.

***Concern on Incremental Contribution***

The problem may seem incremental, as it combines existing methods' strengths, but it addresses the practical challenge of balancing communication cost and performance in FL.


[1]FedBABU: Toward Enhanced Representation for Federated Image Classification, ICLR 2022.

[2]SphereFed: Hyperspherical Federated Learning, ECCV 2022

[3]No Fear of Classifier Biases: Neural Collapse Inspired Federated Learning with Synthetic and Fixed Classifier, ICCV 2023

[4] FedDr+: Stabilizing Dot-regression with Global Feature Distillation for Federated Learning. FedKDD 2024

[5]FedFN: Feature Normalization for Alleviating Data Heterogeneity Problem in Federated Learning. NeurIPS Workshop 2023, Federated Learning in the Age of Foundation Models.

**Questions:**

While I currently have several concerns that have led to a lower score, I am open to increasing the score if these issues are adequately addressed during the rebuttal period.

**Questions and Suggestions:**

The use of pretrained models in federated learning (FL) is a promising and timely research direction, especially given the current trend in the deep learning community toward leveraging foundation models effectively. However, as mentioned earlier in the Related Work section, using a pretrained model is not always superior to using a randomly initialized model. Specifically, findings from FedFN [1], Section 5.2, highlight scenarios where pretrained models can negatively impact global model performance in high heterogeneous settings.

Given this, it would be beneficial for the authors to include experimental comparisons between using a pretrained model and a randomly initialized model. These comparisons should cover various baselines and the proposed algorithm to provide a clearer understanding of whether the pretrained model genuinely improves performance.

Additionally, if the characteristics of the training data used for the pretrained model(e.g. ImageNet) are significantly different from the test data(e.g. SVHN) targeted by the global model, using a pretrained model could potentially be detrimental. It is important to clarify what specific test dataset the final foundation model and redefined classifier are targeting, as this information does not seem to be explicitly stated in the Preliminaries section.


Furthermore, a fundamental challenge in FL is the heterogeneity of client data, which often leads to **class imbalance** issues within each client’s local dataset. However, what differentiates FL from traditional class imbalance problems is the presence of **missing classes**, where certain classes are entirely absent from a client's dataset. This problem is especially pronounced as data heterogeneity increases, causing missing classes to occur more frequently across clients.

The proposed algorithm sends class frequency information from each client, but in the case of missing classes, this would simply convey a value of 0. I am concerned that the algorithm might be particularly vulnerable to the impact of these missing classes. Could the authors explain how their proposed algorithm is designed to mitigate this vulnerability in the context of FL, and why it might still perform well despite the challenges posed by missing classes?


[1]FedFN: Feature Normalization for Alleviating Data Heterogeneity Problem in Federated Learning. NeurIPS Workshop 2023, Federated Learning in the Age of Foundation Models.

---

> ### Author Response · Authors · 2024-11-21
> **Response 1/2**
>
> We thank the Reviewer for their feedback and are encouraged that, despite the extremely low overall score, the Reviewer acknowledges that our approach makes a timely contribution to federated learning with pre-trained model, achieving strong perfomance with minimal additional overhead when compared to the state-of-the-art competitors FedNCM and Fed3R. However, we are surprised by the strong reject recommendation, as it does not seem to fully align with the comments and concerns raised in the review. We are confident that we can adequately address the Reviewer’s concerns in this rebuttal. Below, we provide detailed responses to the issues raised.
>
> >Related Works - the use of fixed classifiers.
>
> We thank the reviewer for suggesting these related papers. While we already discussed [3] in our related work, we have now added the discussion on the impact of freezing classifiers in federated settings with appropriate references [1,2,4] in the updated version of our paper (see L097-100)
>
> > Pretrained models are not always advantageous in federated settings.
>
> In the original manuscript, we discussed several recent published works on federated learning with pre-trained models (Nguyen et al., 2023; Tan et al., 2022b; Chen et al., 2022; Qu et al., 2022; Shysheya et al., 2022; Legate et al., 2023a; Fanı̀ et al., 2024). All of these works show that using pre-trained models significantly benefit federated learning in highly non-iid settings using different federated optimization methods across several datasets (CIFAR-10, CIFAR-100, Stack Overflow, FEMNIST, Reddit, Flowers, CUB, Stanford cars, EuroSAT-Sub, iNaturalist-Users-120K). Our findings further substantiate these observations (see our comparison below with random initialization). As suggested by the Reviewer, we now refer to findings from FedFN [5] which show that in some settings using a pre-trained ResNet-18 model on the CIFAR-10 dataset negatively impacts the learning of the global model. We have added this discussion in L105-107.
>
> >Preliminaries:
>
> We have clarified the following in the revised manuscript:
> - L117: $D_k$ refers to the local dataset.
> - L127: We clarified the loss function. Here, we do not provide details about how the loss is calculated because this is a general federated learning framework which can use different loss functions and employ different ways of computing the loss.
> - L129-130: We now clarify in L131 that the proposed method do not involve any training or local model updates.
>
> >Privacy concerns on sending classwise statistics: The algorithm sends the class-wise frequency of the data held by clients to the central server.  In fact, there are many previous FL papers that have communicated class frequency information and provided justifications. Citing these studies would strengthen the discussion, but this type of content is entirely missing.
>
> We thank the reviewer for pointing out this. Our approach, like other methods we cited (e.g  FedNCM (Legate et al., 2023) and CCVR (Luo et al. 2021)), requires transmitting class-wise frequencies from clients to global server. We agree with the reviewer that this could raise privacy concerns, as sharing class-wise statistics may expose client class distribution. We have updated the paper to explicitly discuss this issue.
>
> Motivated by the Reviewer suggestion (and the suggestion of the Reviewer [6242](https://openreview.net/forum?id=7NtAIghBsE&noteId=qTJF4h1WaX)), we decided to investigate methodologies to mitigate these privacy concerns. We propose perturbing class-wise statistics with different types and intensities of noise before transmitting them to the global server and evaluate the performance of FedCOF. Specifically, we perturb the class-wise statistics as follows:
> $$ñ_{k,c}= \max(n_{k,c} + \sigma^{\text{noise}}_{\epsilon},0).$$
>
> Here $\sigma^{\text{noise}}_{\epsilon}$ represents noise with intensity parametrized by $\epsilon$. The $\max$ operator ensure non-negative values in client statistics.
>
> We vary the intensity of $\epsilon$ and the type of noise applied. We consider Uniform, Gaussian and Laplacian noise. Our findings demonstrate that the performance of FedCOF is robust to the noise type with varying intensities. These results suggest that perturbing statistics can mitigate privacy concerns stemming from the exposure of client class-wise frequencies. We added this empirical analysis in the Supplementary Material (Appendix L).

---

> ### Author Response · Authors · 2024-11-21
> **Response 2/2**
>
> >Concern on Incremental Contribution: The problem may seem incremental, as it combines existing methods' strengths, but it addresses the practical challenge of balancing communication cost and performance in FL.
>
> We thank the Reviewer for recognizing that we address the practical challenge of balancing communication cost and performance in FL. However, we respectfully disagree with the assertion that our proposed approach may appear incremental. In this work we introduce several novel contributions to training-free approaches for Federated Learning with pre-trained models. Firstly, we propose an **unbiased estimator for class covariances** (Proposition 2) that requires only client means, and we mathematically prove this result in the Supplementary Material (Appendix C). Secondly, we establish a **connection between the Ridge Regression solution and class feature covariances** (Proposition 3), and we again mathematically prove this result in the Supplementary Material (Appendix D). The only aspect that may appear incremental is our use of class covariances to initialize a Ridge Regression classifier. However, even in this case, **we propose a different classifier initialization than Fed3R by demonstrating that using between-class scatter matrices decrease performance of the standard Ridge Regression classifier** (see Equation 11). On the basis of these empirical observations, we remove these relationships for classifier initialization and **demonstrate improved performance over Fed3R**.
>
> These contributions are not incremental but are grounded in a careful analysis of existing literature. We provide a novel, mathematically sound, and practically effective solution to address the challenge of training-free methods for pre-trained models in Federated Learning.
>
> > It would be beneficial for the authors to include experimental comparisons between using a pretrained model and a randomly initialized model.
>
> To clarify the impact of using pre-trained models, we conducted additional experiments using a randomly initialized model. Specifically, we conduct these experiments employing SqueezeNet on CIFAR-10 and CIFAR-100. These experiments were conducted in a highly heterogeneous setting in which client data was assigned using a Dirichlet distribution with $\alpha=0.1$, following standard practice. We discuss these details in Appendix H (Impact of using pre-trained models, Table 5) in the revised version of paper. For convenience we report these results in the following table:
> ||||CIFAR10||CIFAR100|
> |:--:|:--:|:--:|:--:|:--:|:--:|
> |Method|Pre-trained|Acc. (↑)|Comm. (in MBs) (↓)|Acc. (↑)|Comm. (in MBs) (↓)|
> |FedAvg|no|37.3|74840|23.9|79248|
> |FedAdam |no|60.5|74840|44.3|79248|
> |FedAvg|yes|84.7|37420|56.7|39624|
> |FedAdam|yes|85.5|37420|62.5|39624|
>
> Our results demonstrate that federated training with a pre-trained SqueezeNet model significantly outperforms a randomly initialized model when using standard methods on CIFAR-10 and CIFAR-100.
>
> >If the characteristics of the training data used for the pretrained model(e.g. ImageNet) are significantly different from the test data(e.g. SVHN) targeted by the global model, using a pretrained model could potentially be detrimental.
>
> The assumption that the pre-training data is relevant to the target dataset is a limitation of all existing works using pre-trained models in federated learning (Nguyen et al., 2023; Tan et al., 2022b; Chen et al., 2022; Qu et al., 2022; Shysheya et al., 2022; Legate et al., 2023a; Fanı̀ et al., 2024) and across other domains. Similar to most existing works, we use weights pre-trained on ImageNet-1k. We also mention in the limitations section of our paper (L537-539), “our method assumes the existence of a pre-trained network. If the domain shift with the client data is sufficiently large, this is expected to impact the performance.”
>
> > The proposed algorithm sends class frequency information from each client, but in the case of missing classes, this would simply convey a value of 0. Could the authors explain how their proposed algorithm is designed to mitigate this vulnerability in the context of FL, and why it might still perform well despite the challenges posed by missing classes?
>
> All of our experiments have missing classes at clients which means that each client contains only a subset of the total classes. This is the nature of federated learning with highly heterogeneous distributions following standard practice of using dirichlet distribution with $\alpha=0.1$. Our proposed method FedCOF requires sharing of class means and class counts from each client only for those classes which are present in the respective clients. For missing classes at each client, we do not send any mean or class count. At the server side, we use means and counts for a particular class $c$ only from those clients which contain class $c$. Thus, our method is not affected by the missing classes phenomenon and is independent of how many classes are there in each client.

---

> > ### Comment · Reviewer_jBiQ · 2024-11-22
> > **Response to authors (1/2)**
> >
> > Dear Authors,
> >
> > Thank you for your detailed response to the comments. While your reply addresses some of my concerns, a few issues remain.
> >
> > **1. Related Work**
> >
> > I noticed that 'FedBabu' was mentioned in L100, but I recommend correcting it to 'FedBABU,' as per the naming convention used in the original paper. Additionally, in the Related Work section, it would be helpful to clearly highlight how your approach differentiates itself from prior work. The current presentation mainly lists previous studies without clearly distinguishing how your work offers unique contributions, particularly within the context of FL with pretrained models.
> >
> > I agree with the growing interest in foundation and pretrained models in deep learning, but I still have concerns about their necessity in federated learning. As I mentioned, a discussion of the drawbacks highlighted by FedFN, particularly in scenarios with data heterogeneity, where applying pretrained models leads to worse performance than training from a randomly initialized model, would be beneficial. Addressing these concerns and explaining how your approach overcomes them would strengthen your argument.
> >
> > Despite these concerns, it’s important to highlight why your approach is meaningful. While FedFN involves local updates and aggregation from local models, your work focuses on training-free FL, which may offer more robustness in heterogeneous environments. Clearly articulating how your method differs and the advantages of these distinctions would further support the significance of your work.
> >
> > **2. Preliminaries**
> >
> > I understand your goal is to utilize the clients' datasets and pretrained models to create a good global model even in situations where the target domain differs from the 'aggregated train dataset,' which is the union of the clients' datasets. However, there is some ambiguity in Section 3.1 on Problem Formulation. Your approach does not involve local updates, so the need for local clients to minimize the loss does not seem necessary. However, Equation (1) in this section could mislead the reader into thinking that local clients need to fit their own datasets. This is misleading and could cause confusion about the overall goal of the paper. I would suggest making the problem formulation more explicit in that context. You also mention the impact of domain differences between the aggregated training dataset and the target domain. It would be useful to clarify that your experiments focus on cases similar to ImageNet-CIFAR, where the domain gap is not very large.
> >
> > **3. Privacy Concerns (Section 4.1 Motivation)**
> >
> > Regarding the privacy concerns with sharing class-wise frequencies, I acknowledge your response in which you address the issue by adding noise to the class frequencies before transmission.
> >
> > However, I understand that your paper primarily focuses on results where the pure (non-noisy) frequencies are exposed, rather than situations where noise is applied to the class frequencies. Given that these experiments focus on pure frequencies in the main table, I believe the justification for using pure frequencies in FL needs to be clearly explained in Section 4.1. The motivation for using pure class frequencies in this context should be adequately addressed, as it is a key aspect of the paper and relates to the core privacy considerations in federated learning.
> >
> > **4. Pretrained Model vs. Randomly Initialized Model**
> >
> > I have reviewed your experiments comparing pretrained models and randomly initialized models. I agree that, in scenarios with less domain gap (e.g., ImageNet-CIFAR), your approach shows promise.
> >
> > ---

---

> > > ### Comment · Reviewer_jBiQ · 2024-11-22
> > > **Response to authors (2/2)**
> > >
> > > **5. Missing Classes Concern**
> > >
> > > The sources of missing classes may arise from two factors: local data heterogeneity and global imbalance within the aggregated train dataset, where major and minor classes may exist. I believe the reason your method performs well in handling missing classes is because the aggregated train dataset in your experiments is class-balanced. However, I think that a more practical situation would involve an unbalanced aggregated train dataset. In such cases, where missing classes result from global minor factors, could your approach still perform well?
> > >
> > > **6. Industry Implementation Perspective**
> > >
> > > You mentioned that pretrained models in FL work well when the gap between the aggregated train dataset and target domain is small. I agree with this point.
> > >
> > > However, in real-world FL environments, the client's dataset is typically private, meaning that the class balance in the aggregated dataset is unknown, and we cannot assess the gap between the aggregated train dataset and the target domain.
> > >
> > > Based on your argument, it seems that your approach is effective only when the gap between the aggregated train dataset and the target domain is small. If that is the case, I personally believe the value of your work could be relatively limited, as this would make the method less applicable in real-world scenarios with potentially large domain gaps.
> > >
> > > **7. Novelty Concern**
> > >
> > > I still think that, in terms of novelty, this work can be seen as a method that builds upon the existing strengths of similar approaches in the field. While it does offer a valuable contribution, I consider the novelty to be relatively low as it primarily adapts methods used in existing research, rather than introducing a fundamentally new approach.

---

> > > > ### Author Response · Authors · 2024-11-24
> > > > **Response (1/4)**
> > > >
> > > > We thank the reviewer for taking time and clarifying remaining doubts. We answered all the reviewers minor concerns adequately in our previous response, and honestly think the remaining reviewer's criticisms in no way reflect their very low score.
> > > >
> > > > The scoring of papers is a vital aspect of the reviewing process: reviewers are asked to justly balance strong and weak points of a paper in arriving at their recommendation. We are hopeful that our response will clarify the remaining minor concerns.
> > > >
> > > > > 1. Related Work: I noticed that 'FedBabu' was mentioned in L100, but I recommend correcting it to 'FedBABU,' as per the naming convention used in the original paper. Additionally, in the Related Work section, it would be helpful to clearly highlight how your approach differentiates itself from prior work. The current presentation mainly lists previous studies without clearly distinguishing how your work offers unique contributions, particularly within the context of FL with pretrained models. I agree with the growing interest in foundation and pretrained models in deep learning, but I still have concerns about their necessity in federated learning. As I mentioned, a discussion of the drawbacks highlighted by FedFN, particularly in scenarios with data heterogeneity, where applying pretrained models leads to worse performance than training from a randomly initialized model, would be beneficial. Addressing these concerns and explaining how your approach overcomes them would strengthen your argument. Despite these concerns, it’s important to highlight why your approach is meaningful. While FedFN involves local updates and aggregation from local models, your work focuses on training-free FL, which may offer more robustness in heterogeneous environments. Clearly articulating how your method differs and the advantages of these distinctions would further support the significance of your work.
> > > >
> > > > - We will correct FedBabu to FedBABU in the final version.
> > > > - We discuss the significance and motivation of our work in the context of most relevant works in the introduction L044-084. We believe that it is clear from the introduction how our work is different from existing relevant works. We will add a statement in the related work section to highlight our contribution.
> > > > - We make no claim about the *necessity* of pre-trained models for federated learning. We respect the reviewer opinion regarding their doubt about the necessity of pre-trained models for federated learning. However, as we said in our previous response, the positive impact of pre-trained models has already been established by several papers published in NeurIPS, ICLR, CVPR, ICML in recent years (Nguyen et al., 2023; Tan et al., 2022b; Chen et al., 2022; Qu et al., 2022; Shysheya et al., 2022; Legate et al., 2023a; Fanı̀ et al., 2024). These papers thoroughly discuss the dramatic improvement in performance using pre-trained models across several datasets for federated learning. Furthermore, the observations on a single small dataset (CIFAR-10) in FedFN [5] paper are far from conclusive. For instance, from the results in the FedFN paper, FedBABU achieves better performance using pre-trained model (49.78 over 49.21) in the most heterogeneous setting (s=2) but the randomly initialized model achieves higher accuracy in less heterogeneous settings (s=3, s=5) and again the pre-trained model performs better in the least heterogeneous setting (s=10). The FedFN paper does not explain anywhere in the paper why this happens and provides no insights other than one sentence stating the experimental results. While this needs to be investigated further in future works, we believe it is very little evidence coming to the general conclusion that using pre-trained models is not a better choice. As requested by the reviewer, we already mention this briefly and cite [5] in our related work and we believe more discussion of this is beyond the scope of our work.

---

> > > > ### Author Response · Authors · 2024-11-24
> > > > **Response (2/4)**
> > > >
> > > > > 2. Preliminaries: I understand your goal is to utilize the clients' datasets and pretrained models to create a good global model even in situations where the target domain differs from the 'aggregated train dataset,' which is the union of the clients' datasets. However, there is some ambiguity in Section 3.1 on Problem Formulation. Your approach does not involve local updates, so the need for local clients to minimize the loss does not seem necessary. However, Equation (1) in this section could mislead the reader into thinking that local clients need to fit their own datasets. This is misleading and could cause confusion about the overall goal of the paper. I would suggest making the problem formulation more explicit in that context. You also mention the impact of domain differences between the aggregated training dataset and the target domain. It would be useful to clarify that your experiments focus on cases similar to ImageNet-CIFAR, where the domain gap is not very large.
> > > >
> > > > - The preliminaries section is meant to introduce and formalize the general setting and standard practice in federated learning works. In section 3.1, we briefly explain the federated learning problem (as we already state in L113) and we do not say anything about our proposed method. This is to introduce the FL problem. In section 3.2, we introduce the most relevant works of FedNCM and Fed3R which are important to understand the context of our work. Following the formalization of FL problem and training-free methods in section 3, we discuss our method in details in section 4 and also provide algorithm 1 in page 7 to clarify the exact steps of our method.
> > > > - We mention this in L130-131, where we clearly state that we do not perform local updates and use a frozen model. We believe this should remove any confusion. We also use federated training for the finetuning and linear probing experiments in section 5.2. So, the definition of federated learning we provide in the preliminaries section is important for understanding the paper.
> > > > - The reviewer's claim that our experiments focus on cases similar to ImageNet-CIFAR, in which the domain gap is not very large, is not true. We have tested our method on 5 datasets, not just on CIFAR-100. For instance, we perform experiments with ImageNet-R [Hendrycks et al.] which is an out-of-distribution dataset and proposed to evaluate out-of-distribution generalization using ImageNet pre-trained weights. It contains data with multiple styles like cartoon, graffiti and origami which is not seen during pre-training. We also consider fine-grained datasets like CARS and CUB200. Notably, we also use iNaturalist-Users-120k dataset in our experiments, which is a real-world, large-scale dataset proposed by [Hsu et al.] for federated learning and contains 120k training images of natural species taken by citizen scientists around the world, belonging to 1203 classes spread across 9275 clients. We will add this discussion in the final version of the paper.
> > > >
> > > > > 3. Privacy Concerns (Section 4.1 Motivation): Regarding the privacy concerns with sharing class-wise frequencies, I acknowledge your response in which you address the issue by adding noise to the class frequencies before transmission. However, I understand that your paper primarily focuses on results where the pure (non-noisy) frequencies are exposed, rather than situations where noise is applied to the class frequencies. Given that these experiments focus on pure frequencies in the main table, I believe the justification for using pure frequencies in FL needs to be clearly explained in Section 4.1. The motivation for using pure class frequencies in this context should be adequately addressed, as it is a key aspect of the paper and relates to the core privacy considerations in federated learning.
> > > >
> > > > - All our results are based on the communication of pure class frequencies since this only raises minor privacy concerns compared to the communication of full covariances.  We will add the following footnote in the privacy concerns section 4.1: "Our method does require communication of class frequencies which could raise privacy concerns; in Appendix L we perform an extensive evaluation of this." Also, we have mentioned in line 283 after our proposed approach which points to the supplementary material where we provide an in-depth discussion and analysis of the potential privacy concerns arising from the exposure of class frequency statistics.

---

> > > > ### Author Response · Authors · 2024-11-24
> > > > **Response (3/4)**
> > > >
> > > > > 4. Pretrained Model vs. Randomly Initialized Model. I have reviewed your experiments comparing pretrained models and randomly initialized models. I agree that, in scenarios with less domain gap (e.g., ImageNet-CIFAR), your approach shows promise.
> > > >
> > > > - In the experiments asked by the reviewer, we only show the impact of using pre-trained models. In that experiments, we used a very similar setting of FedFN: pre-trained network and target dataset CIFAR-10 and CIFAR-100 and high hetereogenous distribution. Our results show that for instance with Squeezenet the pre-training significantly improves performance. These results does not aim to show that pre-training is always better than random initialization but rather to demonstrate to the reviewer that the findings from FedFN are not generalizable, remain very preliminary, and require further experimental investigation.
> > > > - These results do not say anything about our approach. With respect to other training-free method such has FedNCM and Fed3R, our method show good performance on several dataset with larger domain shifts from ImageNet like ImageNet-R, iNaturalist-Users-120k as discussed above in response to questions about the preliminaries section.
> > > >
> > > > > 5. Missing Classes Concern. The sources of missing classes may arise from two factors: local data heterogeneity and global imbalance within the aggregated train dataset, where major and minor classes may exist. I believe the reason your method performs well in handling missing classes is because the aggregated train dataset in your experiments is class-balanced. However, I think that a more practical situation would involve an unbalanced aggregated train dataset. In such cases, where missing classes result from global minor factors, could your approach still perform well?
> > > >
> > > > - Our method is robust to aggregated class imbalance. We normalize the classifier weights (last step of FedCOF on server side) to account for the class imbalance at the server level after aggregation. We discuss this in L353-355 and also in Algorithm 1.
> > > > - The reviewer's statement that the aggregated train dataset in our experiments is class-balanced is not true. Although most existing datasets used for federated learning have a balanced aggregated dataset, we consider class-imblanced datasets like ImageNet-R and CARS, and our method performs very well in class-imbalanced conditions as well. Our method performs well in all situations of missing classes (from both local data heterogeneity and global imbalance). We would also like to highlight that existing training-free methods like FedNCM and Fed3R also work in all the missing classes situations and this is not a concern.
> > > >
> > > > Class imbalance for first 30 classes in ImageNet-R:
> > > > |Index|0|1|2|3|4|5|6|7|8|9|10|11|12|13|14|15|16|17|18|19|20|21|22|23|24|25|26|27|28|29|
> > > > |-----|-|-|-|-|-|-|-|-|-|-|--|--|--|--|--|--|--|--|--|--|--|--|--|--|--|--|--|--|--|--|
> > > > |Value|184|160|154|81|181|142|139|44|194|145|69|136|165|56|155|80|175|115|83|137|101|270|280|150|96|64|259|237|206|74|
> > > >
> > > > Class imbalance for first 30 classes in CARS:
> > > > |Index|0|1|2|3|4|5|6|7|8|9|10|11|12|13|14|15|16|17|18|19|20|21|22|23|24|25|26|27|28|29|
> > > > |-----|-|-|-|-|-|-|-|-|-|-|--|--|--|--|--|--|--|--|--|--|--|--|--|--|--|--|--|--|--|--|
> > > > |Value|45|32|43|42|41|45|39|45|41|33|38|37|41|43|43|44|41|43|41|46|42|43|40|45|40|34|36|41|43|42|
> > > >
> > > > >6. Industry Implementation Perspective. You mentioned that pretrained models in FL work well when the gap between the aggregated train dataset and target domain is small. I agree with this point. However, in real-world FL environments, the client's dataset is typically private, meaning that the class balance in the aggregated dataset is unknown, and we cannot assess the gap between the aggregated train dataset and the target domain.
> > > > Based on your argument, it seems that your approach is effective only when the gap between the aggregated train dataset and the target domain is small. If that is the case, I personally believe the value of your work could be relatively limited, as this would make the method less applicable in real-world scenarios with potentially large domain gaps.
> > > >
> > > > - We disagree with the reviewer that pre-trained networks are of limited importance for federated learning. We do think pretrained models will play an important role in the future of federated learning, especially in industrial contexts. Please refer to our above response where we discuss the excellent performance of our approach on large-scale datasets with large domain gaps (ImageNet-R, iNaturalist-Users-125k).

---

> > > > ### Author Response · Authors · 2024-11-24
> > > > **Response (4/4)**
> > > >
> > > > >7. Novelty Concern. I still think that, in terms of novelty, this work can be seen as a method that builds upon the existing strengths of similar approaches in the field. While it does offer a valuable contribution, I consider the novelty to be relatively low as it primarily adapts methods used in existing research, rather than introducing a fundamentally new approach.
> > > >
> > > > - FedCOF is *not* an adaptation of existing works. We have rigorously derived an unbiased estimator of class covariances using only first-order statistics, thus allowing us to exploit second-order statistics while incurring the same communication costs as FedNCM. Moreover, we again rigorously derive the second-order Fed3R classifier in terms of class covariances and demonstrate how it can be *significantly* improved by eliminating cross-class contributions in their formulation. Thus we do not simply propose a new method that demonstrates state-of-the-art empirical results, but have rigorously and mathematically proved *why this is the case*. Every research paper builds on the context of existing works. We believe that the primary problem we solve in this paper -- estimating covariances from only means is novel and has not been discussed or attempted in any existing work. If the reviewer insists on continuing to doubt the novelty of our contribution, we respectfully ask that they justify exactly *how* our work is a derivative adaptation of existing works, and specifically cite *which* works.

---

> ### Comment · Reviewer_jBiQ · 2024-11-24
> **Request and Question of Response (1/4-2/4)**
>
> Dear Authors,
>
> Thank you for your detailed response to the comments. First of all, I have increased the score to 3, as you have addressed some of my concerns. Based on the points you mentioned, I kindly ask you to finalize the draft, as I need to evaluate the version that has been reflected so far, rather than future versions.
>
> I do have one question regarding the methodology. In this study, the main tables were all based on experiments using pure class frequencies. Since I believe this is not the first Federated Learning study to utilize pure class frequencies, I would appreciate further clarification on this matter. Specifically, in the text, the justification for using feature prototypes is explained as: "Sharing only class means provides a higher level of data privacy compared to sharing raw data, as prototypes represent the mean of feature representations. It is not easy to reconstruct exact images from prototypes with feature inversion attacks, as shown by (Luo et al., 2021)." Given this, is there a comparable justification for using pure class frequencies in the context of Federated Learning?

---

> ### Comment · Reviewer_jBiQ · 2024-11-24
> **Request and Question of Response (3/4-4/4)**
>
> Dear Authors,
>
> Thank you for your detailed response to the comments. However, it seems that the concern I raised below has not been fully addressed. I would appreciate further clarification on this matter.
>
> The reason I raised a question regarding the problem setting is that I felt the rationale for introducing a pretrained model into Federated Learning was not clear.
>
> As I mentioned earlier, my question is whether using a pretrained model is truly better than random initialization in arbitrary scenarios (arbitrary domain gap, arbitrary heterogeneity).
>
> I believe this is important because, in actual Federated Learning, client datasets are not publicly available. Therefore, in such arbitrary situations, using a pretrained model should not be worse than using a randomly initialized model.
>
> From what I understand, the study demonstrates superior performance in situations with domain gaps compared to other algorithms, but I don’t believe it provides sufficient justification for using pretrained models comparing to randomly initialized model in such arbitrary situations.

---

> > ### Author Response · Authors · 2024-11-25
> >
> > We thank the reviewer for discussions and we have updated the paper now with the previously discussed changes.
> >
> > > I do have one question regarding the methodology. In this study, the main tables were all based on experiments using pure class frequencies. Since I believe this is not the first Federated Learning study to utilize pure class frequencies, I would appreciate further clarification on this matter. Specifically, in the text, the justification for using feature prototypes is explained as: "Sharing only class means provides a higher level of data privacy compared to sharing raw data, as prototypes represent the mean of feature representations. It is not easy to reconstruct exact images from prototypes with feature inversion attacks, as shown by (Luo et al., 2021)." Given this, is there a comparable justification for using pure class frequencies in the context of Federated Learning?
> >
> > We want to stress that the primary goal of our work is to reduce communication costs of sharing high-dimensional covariances but still obtain the performance gain of using client covariance statistics. An additional feature of our method is that it adds more security since the covariances or feature relationships can leak very sensitive client information. Sharing class frequencies is a very minor concern compared to sharing entire covariance matrices.
> >
> > The comparable justification would be: "Following (Legate et al., 2023a; Luo et al., 2021), we use class frequencies from clients since it only quantifies the client data while not revealing any information at the data or feature level." We add this in L206-207.
> >
> > >The reason I raised a question regarding the problem setting is that I felt the rationale for introducing a pretrained model into Federated Learning was not clear.
> > As I mentioned earlier, my question is whether using a pretrained model is truly better than random initialization in arbitrary scenarios (arbitrary domain gap, arbitrary heterogeneity).
> > I believe this is important because, in actual Federated Learning, client datasets are not publicly available. Therefore, in such arbitrary situations, using a pretrained model should not be worse than using a randomly initialized model.
> > From what I understand, the study demonstrates superior performance in situations with domain gaps compared to other algorithms, but I don’t believe it provides sufficient justification for using pretrained models comparing to randomly initialized model in such arbitrary situations.
> >
> > The reviewer is right that it is possible to imagine datasets for which pretrained models do not provide additional performance gain. However, on most datasets, even with large domain shifts, used by the community to evaluate federated learning, we (and others [X, Y]) found that pretrained models provide a significant performance advantage. For instance, FedNCM [X] shows in figure 1 and in the Appendix that random initialization does not achieve good performance even after too many training rounds. Furthermore, we would like to refer the reviewer to paper [Y] which explicitly talks about why pre-training is helpful for Federated Learning and provides in-depth empirical and theoretical discussions on several aspects. With this paper, we do not want to advocate abandoning research on federated learning from scratch, and we hope the community will continue working on both theory of federated learning starting from scratch and from pretrained models.
> >
> > [X] Legate et al., Guiding the last layer in federated learning with pre-trained models. In Advances in Neural Information Processing Systems, 2023.
> >
> > [Y] Nguyen et al., Where to begin? exploring the impact of pre-training and initialization in federated learning. In The Eleventh International Conference on Learning Representations, 2023.

---

> > > ### Comment · Reviewer_jBiQ · 2024-11-27
> > >
> > > Dear Authors,
> > >
> > > Thank you for your response to my concerns.
> > >
> > > First of all, I still have lingering doubts regarding the **problem justification (why pretrained models should be applied to Federated Learning)** and the **novelty scope**.
> > >
> > > Regarding the novelty scope, I set that aside for now, but the primary issue remains the problem justification. From an industry perspective, I still question whether it really makes sense to introduce a pretrained model into Federated Learning compared to a random initialized model. This is particularly concerning because introducing a pretrained model comes with significant costs. I believe this concern is especially valid when there is a **large domain gap**.
> > >
> > > Therefore, I think it is important to strengthen the justification by analyzing the tendencies of both the proposed algorithm and the baseline algorithm in scenarios with large domain gaps.
> > >
> > > However, you have defended this by referencing other papers.
> > >
> > > The paper [X] appears to be the motivation behind your study. Since this paper served as the motivation, is there any reason why the follow-up study (your paper) would not experiment with random initialization vs pretrained models in a general scenario? I am concerned about this.
> > >
> > > Additionally, the paper [Y] pertains to a setting where local updates are allowed in Federated Learning, which is different from the train-free setting in your study. Therefore, the results in [Y] do not necessarily apply here. In fact, Section 6 of paper [Y] (Motivation) states: **When evaluating FL algorithms, researchers should experiment with both pre-trained (if available) and random weights, as the initialization can clearly impact the relative performance of different methods, and both initialization schemes may be relevant to practice.**
> > >
> > > **For these reasons, I believe that the results using random initialization should also be reported in the main table, and the justification for introducing pretrained models, which your study emphasizes, should be clearly discussed in the main text rather than in the appendix.**
> > >
> > > While I have raised concerns about the problem justification multiple times, I feel that these concerns have not been fully addressed, and as such, I am unable to increase the score at this time.

---

> > > > ### Author Response · Authors · 2024-12-01
> > > >
> > > > We thank the reviewer for discussions but we do not agree with the reviewer's position that we should demonstrate the benefits of pre-trained models in **arbitrary scenarios**. We have already conducted experiments comparing pre-trained versus random initializations in Table 5 (Appendix H) and our main paper contains sufficient discussion and references on "justification for introducing pretrained models" in L012-013, L044-047, L100-106. Our contribution focuses on *training-free* federated learning -- a scenario in which starting from pre-trained models is clearly beneficial. Prior work (FedNCM [a]) has established the benefits of starting from pre-trained models, and *our* extensive experiments below also confirm this:
> > > >
> > > > |Method|Pre-trained|CIFAR100|CIFAR100|ImageNetR|ImageNetR|CUB200|CUB200|CARS|CARS|
> > > > |:--:|:--:|:--:|:--:|:--:|:--:|:--:|:--:|:--:|:--:|
> > > > | | |Acc.(↑)|Comm.(MB)(↓)|Acc.(↑)|Comm.(MB)(↓)|Acc.(↑)|Comm.(MB)(↓)|Acc.(↑)|Comm.(MB)(↓)|
> > > > |FedAvg (1k rounds)|no|37.2|198120|1.2|210420|3.9|210420|3.3|209940|
> > > > |FedAdam (1k rounds)|no|44.4|198120|2.1|210420|12.2|210420|7.8|209940|
> > > > |FedCOF (ours, 1 round)|yes|56.1|5.9|37.8|7.1|53.7|4.8|44.0|5.4|
> > > >
> > > > Even after federated training for 1000 rounds, random initialization with squeezenet performs very poorly in difficult datasets like ImageNet-R, CARS and CUB200. We will add these results in the Appendix with all implementation details.
> > > >
> > > > Given our already extensive evaluation on multiple datasets -- including large-scale datasets (especially ImageNet-R with large domain gap as we explained in our previous response) -- and that *all* of our experiments consider high heterogeneity across clients, **searching for additional scenarios with "arbitrary domain gap, arbitrary heterogenity" where the pre-trained model may be worse than random initialization is beyond the scope of our work**. The focus of this work is on *training-free methods* for federated learning, which by very definition require a pre-trained network.
> > > >
> > > > [a] Legate et al., Guiding the last layer in federated learning with pre-trained models. In Advances in Neural Information Processing Systems, 2023.

---

### Author Response · Authors · 2024-11-21
**Global Response**

We thank all reviewers for their insightful feedback aimed at improving the quality of our work. The reviewers agree that the paper is well-written (**Wumb**), address a relevant and interesting problem (**z1Dd**), presents a timely contribution (**jBiQ**), with clear and easy to understand derivations (**Wumb**), propose a sound (**z1Dd**) and novel (**6242**,**z1Dd**) covariance estimator with theoretical guarantees (**6242**) and provide comprehensive (**z1Dd**) and extensive (**6242**) empirical evaluation.

Here we provide a summary of our responses and highlight new results presented during the discussion period. We have updated the paper with all the experiments detailed below and highlighted the additional clarifications.

In response to the concerns raised by reviewer [jBiQ](https://openreview.net/forum?id=7NtAIghBsE&noteId=5ZSh899J9u), we updated the related works by citing FedBABU, SphereFed, FedDr+, neural collapse and FedFN. We also updated the preliminaries section to clarify notations. We performed experiments to show that using pre-trained networks for FedAvg and FedAdam significantly outperforms training with a randomly initialized network. We have updated the paper to discuss the privacy concerns raised by sharing class statistics and address that using perturbations. We also clarify that our method is not affected by the missing classes problem.

In response to the concerns raised by reviewer [Wumb](https://openreview.net/forum?id=7NtAIghBsE&noteId=NcBpn5ddrg) we updated the discussion of FedCOF with multiple rounds in the paper for more clarity. We also discuss the role of the ridge regression parameter and highlight the difference between the Fed3R and FedCOF-Oracle initializations.

In response to the concerns raised by reviewer [z1Dd](https://openreview.net/forum?id=7NtAIghBsE&noteId=vWoj55YR9j), we theoretically analyze the bias of the proposed estimator in non-iid settings with locally different covariances. We also performed experiments on feature-shift settings using DomainNet. Finally, we performed experiments using ResNet-18 to compare with classical federated learning methods and to also compare our method with federated training of linear models. We have also clarified the difference between FedCOF from Fed3R initialization.

In response to the concerns raised by reviewer [6242](https://openreview.net/forum?id=7NtAIghBsE&noteId=qTJF4h1WaX), we now discuss the privacy concerns related to sharing local class frequencies and have proposed a perturbation strategy to address those concerns.

---

> ### Author Response · Authors · 2024-12-03
>
> We thank all reviewers for their valuable feedback and for actively engaging in the discussions. We believe that we have addressed all the concerns of reviewers **Wumb**, **z1Dd**, **6242** and thank them for their positive ratings of our work. We are very thankful to reviewers **z1Dd, 6242** for acknowledging our efforts in the rebuttal phase.
>
> We believe we have thoroughly addressed all concerns raised by reviewer **jBiQ** -- including related work, preliminaries, privacy discussions, missing classes, random initialization, and novelty clarification -- and have updated our paper as a result. Unfortunately the reviewer still recommmends a 'reject' rating, referring to '*lingering doubts*' on the use of pre-trained models and incremental novelty concerns. To summarize our position on these remaining issues:
>
> 1 - **Problem justification (why pretrained models should be applied to Federated Learning)**: We first of all underscore that the focus of our contribution is on *training-free federated learning*, which is a scenario in which the benefits of starting from pre-trained models is evident and firmly established by recent prior works on training-free federated learning [a, b]. Nonetheless -- and contrary to reviewer claims -- our main paper contains ample discussion and references on "justification for introducing pretrained models" in L012-013, L044-047, L100-106.
>
> The reviewer's other concern is on including the random initialization results in the main table of the paper. We have already conducted experiments comparing pretrained versus random initializations in Table 5 (Appendix H) and also in the last response after the pdf update deadline. Since this is not the main focus of our work and the performance using random initialization is *very bad* even after several rounds of training (as also established by FedNCM [a]), we have therefore decided that we will keep these results in the Appendix in any future version.
>
> The reviewer also would like us to investigate whether pre-trained models are useful in "*arbitrary scenarios (arbitrary domain gap, arbitrary heterogeneity)*" but does not provide any specific references. From the literature we know that pre-trained features can be effective even with large domain gaps: pretrained ImageNet features have been applied to several fine-grained datasets [c], medical datasets [d], remote sensing [e], to name just a few. Our experimental evaluation *already* considers datasets with significant domain gaps like ImageNet-R and heterogeneity -- as we have pointed out in our rebuttal.
>
> 2 - **Novelty Concerns**: While other reviewers appreciated the novelty of our work and despite our clarification of novelty twice in the discussion phase, the reviewer did not engage with our clarifications, and has at no point during the review and rebuttal process engaged with the technical content and main contributions of our work.
>
>
> [a] Legate et al., Guiding the last layer in federated learning with pre-trained models. In Advances in Neural Information Processing Systems, 2023.
>
> [b] Eros Fani, Raffaello Camoriano, Barbara Caputo, and Marco Ciccone. Accelerating heterogeneous federated learning with closed-form classifiers. Proceedings of the International Conference on Machine Learning, 2024.
>
> [c] Kornblith, Simon, et al. "Do better imagenet models transfer better?." Proceedings of the IEEE/CVF conference on computer vision and pattern recognition. 2019.
>
> [d] Dack, Ethan, et al. "An empirical analysis for zero-shot multi-label classification on covid-19 ct scans and uncurated reports." Proceedings of the IEEE/CVF International Conference on Computer Vision. 2023.
>
> [e] Corley, Isaac, et al. "Revisiting pre-trained remote sensing model benchmarks: resizing and normalization matters." Proceedings of the IEEE/CVF Conference on Computer Vision and Pattern Recognition. 2024.

---

### Meta-Review · Area_Chair_91Dz · 2024-12-21

**Metareview:**

This paper proposes an interesting train-free federated learning method which leverages (publicly available) pre-trained models to boost performance. The empirical results appear promising.

There has been multiple extensive discussions regarding the impact of the paper's contribution between authors and reviewers as well as between reviewers and AC. During the reviewer-author discussion, there is a debate among the reviewer on the benefit of initializing federated learning with pre-trained models. While this feels at first counter intuitive that initializing federated learning with a pre-trained model might not always help, this is empirically true in cases where there is significant distribution gap between local datasets and/or when local datasets are highly skewed/imbalanced in different ways -- see [*].

[*] https://openreview.net/pdf?id=nw6ANsC66

Otherwise, in standard settings where there is no such imbalance or significant distribution gap across local datasets, there is a verified consensus that initializing federated learning with pre-trained models will help. Upon extensive discussion with the reviewers, I believe the focus of this paper is on the standard setting so the point raised by reviewer jBiQ does not affect the contribution of the paper. The authors however are encouraged to provide extra discussion around this point in their revised paper.

Note that [*] is cited here to reconcile the seemingly opposite points raised by the reviewers during the discussion. The paper is not penalized for not citing/comparing with this recent work (which also focuses on an orthogonal setting).

--

Having said that, the real concern here, however, is that using pre-trained backbone is not a new practice and so this paper should have compared its method with more direct baselines. For example, if we view local models as fine-tuned versions of the large model, we could easily apply existing FL methods (e.g., FedAvg, FedProx etc.) to aggregate the fine-tuning parameters -- see the baselines used in [*].

The authors should have then compared the performance of their proposed (one-shot) method with both multiple- and single-shot variants of those baselines to conclusively demonstrate its benefit. I would expect to see that the performance of the proposed method coming close to or even exceed the performance of the multiple-shot variants while incurring much less communication cost. I believe this is the main point here which needs to be demonstrated more thoroughly.

Appendix H + the main-text experiments currently fall short of achieving this.

--

Overall, I feel that this paper is somewhat below the acceptance bar mainly due to the aforementioned issue with its empirical studies. Otherwise, I agree that its technical idea (pending thorough impact assessment) is sufficiently novel. It essentially boils down to whether the contribution of this paper outweigh its flaws. I have asked the positive reviewers to see if anyone is willing to champion this paper given the above assessment. But, unfortunately, there is no indication that anyone is willing to do so and this paper remains in the borderline.

--

Regardless of the final decision of the PC, I hope the authors would seriously revise the paper to take into account all the key discussion points that I summarize above.

**Additional Comments On Reviewer Discussion:**

Both the AC-reviewer and author-reviewer discussions are very active. A key debate point that arises is whether there is a clear benefit regarding initializing federated learning with pre-trained models.

One reviewer raises a seemingly counter-intuitive point that there might be cases where doing so is worse than going with a random initialization. Upon further debate among the AC and reviewers, we come to an agreement that this can be the case if the distribution gaps across datasets are significant. The AC also pointed out a recent work that shows initializing federated learning with pre-trained model might result in poor performance if local datasets are imbalanced or skewed in different ways.

[*] https://openreview.net/pdf?id=nw6ANsC66

But, the paper's focus is not on such extreme setting so we think that the authors only need to expand a detailed discussion around this point to correctly provide the full (empirical) picture surrounding the use of pre-trained models in federated learning. This is definitely not a show-stopper for this paper.

However, as pointed out in the main meta review, the AC also sees that the real concern here, however, is that using pre-trained backbone is not a new practice and so this paper should have compared its method with more direct baselines. For example, if we view local models as fine-tuned versions of the large model, we could easily apply existing FL methods (e.g., FedAvg, FedProx etc.) to aggregate the fine-tuning parameters -- see the baselines used in [*].

The authors should have then compared the performance of their proposed (one-shot) method with both multiple- and single-shot variants of those baselines to conclusively demonstrate its train-free benefit.

--

Overall, I feel that this paper is somewhat below the acceptance bar mainly due to the aforementioned issue with its empirical studies. Otherwise, I agree that its technical idea (pending thorough impact assessment) is sufficiently novel.

---

### Decision · Program_Chairs · 2025-01-22

Reject